# AI-ASSISTED GENERATION OF DIFFICULT MATH QUESTIONS

## ABSTRACT

Current LLM training positions mathematical reasoning as a core capability. With publicly available sources fully tapped, there is an unmet demand for diverse and challenging mathematics questions. Relying solely on human experts is both time-consuming and costly, while LLM-generated questions often lack the requisite diversity and difficulty. We present a design framework that combines the strengths of LLMs with a human-in-the-loop approach to generate a diverse array of challenging math questions. Initially, leveraging LLM metacognition skills (Didolkar et al., 2024), a strong LLM is used to extract core "skills" from existing math datasets. These skills serve as the basis for generating novel and difficult questions by prompting the LLM with random pairs of core skills that must be utilized in the question. The use of two very different skills within each question makes finding such questions an "out of distribution" task for both LLMs and humans. Our pipeline employs LLMs to iteratively generate and refine questions and solutions through multi-turn prompting. Human annotators then verify and further refine the questions, with their efficiency enhanced via further LLM interactions. Applying this pipeline on skills extracted from MATH dataset (Hendrycks et al., 2021) resulted in **MATH**$^2$ - a dataset of higher quality math questions, as evidenced by: (a) Lower performance of all models on MATH$^2$ than on MATH (b) Higher performance on MATH when using MATH$^2$ questions as in-context examples. Although focused on mathematics, our methodology seems applicable to other domains requiring structured reasoning, and potentially as a component of *scalable oversight*. Also of interest is a striking relationship observed between models' performance on the new dataset: the success rate on MATH$^2$ is the square on MATH. This suggests that successfully solving the question in MATH$^2$ requires a nontrivial combination of two distinct math skills.

## 1 INTRODUCTION

Significant improvement in the capabilities of LLMs (Chowdhery et al., 2023; Anil et al., 2023; Team, 2023; Team et al., 2023; Abdin et al., 2024; Achiam et al., 2023; Touvron et al., 2023) to understand and generate complex mathematical content has been achieved by leveraging all the public data and a fair bit of private data. Sources of high-quality, varied, and difficult mathematical questions are drying up. Even finding new questions for evaluation is getting difficult since newly-released human exams are somewhat similar to past exams, which are potentially present in the LLMs' training datasets. Hence, there is a pressing need for innovative methods to create new, diverse, and challenging questions.

Expert mathematicians and educators possess the deep understanding required to create questions that not only test a wide range of skills but also push the boundaries of what the learners, and by extension, the models, can handle. However, relying solely on human experts is not scalable. Generating synthetic questions using LLMs is feasible at scale (Trinh et al., 2024; Li et al., 2024; Gunasekar et al., 2023; Patel et al., 2024; Toshniwal et al., 2024; Gupta et al., 2023; Lu et al., 2024; Honovich et al., 2022), but often falls short in terms of the necessary difficulty. Huang et al. (2024) employs a similar approach as ours where they extract *topics* and corresponding *keypoints* from a set of seed problems using GPT-4, and then combine the *topic* to generate new questions, again using GPT-4). However, the generated data is meant to be used for the finetuning of models as compared to serving as an evaluation set in our case. As a result, the questions generated in Huang et al. (2024) are

not sufficiently difficult. Similarly, limited work exists on ensuring the necessary diversity in the generated synthetic data. Chan et al. (2024) proposes prompting frontier models to generate questions where each question is generated in the context of a *persona* as a way of ensuring diversity. They use 1M different personas to generate questions, which are then used for finetuning models, leading to significant improvements. This dichotomy between the quality of human-generated questions and the scalability of LLM-generated questions presents a significant challenge (Yu et al., 2024).

## 1.1 EVALUATION SATURATION PHENOMENON

LLM evaluations getting saturated is a well-known issue. Some of the saturation is driven by across-the-board improvements arising from better training and more extensive/better datasets. But a lot has to do with evaluation-specific enhancements that optimize model performance on standard evaluations through techniques like supervised fine-tuning (SFT) on synthetic question-answer pairs. These synthetic pairs can be generated by leading proprietary models when provided with a few examples from the dataset or by filtering the model's own responses (Yue et al., 2023; Yu et al., 2023). Such methods can dramatically boost performance; for example, just 1 million synthetic examples can elevate Llama2 7B's performance on the MATH dataset to levels comparable to GPT-4 (Li et al., 2024).

The distinction between general and evaluation-specific improvements is crucial. The latter may lead to overfitting to particular evaluations rather than a genuine acquisition of mathematical skills. This issue was highlighted when a new version of the GSM8K dataset revealed performance drops in many models, indicating overfitting to the previous dataset version (Zhang et al., 2024). Similarly, leading LLMs performed significantly worse on newer versions of the Chinese GaoKao exam compared to older exams, raising fundamental questions about the depth of their mathematical understanding.

## 1.2 PROPOSED FRAMEWORK: AI-ASSISTED GENERATION OF DIFFICULT MATH QUESTIONS

At first glance, it may seem counterintuitive to use an AI model to generate and correct novel questions that it is unable to solve itself. However, recent research (Arora & Goyal, 2023; Didolkar et al., 2024) demonstrated that top LLMs possess a robust understanding of mathematical skills, including the capability to identify the skills required to solve given questions (Reid et al., 2024; Achiam et al., 2023). This naturally raises the question: *can LLMs operate in the reverse direction, i.e., generate math problems when given a list of skills that have to be tested?* Our initial attempts yielded mixed results. While leading models could produce creative math questions when provided with a list of skills, the majority of these questions exhibited one or more of the following shortcomings: too similar to existing questions in datasets; have errors or nonsensical elements; are too tedious or mechanical to be engaging for human annotators. (See Section 4.) Moreover, they often conflate "difficulty" with tedious calculations, which actually would play to the strength of machines to leverage external tools such as calculators or Python interpreters.

Nevertheless, there were promising instances where LLMs generated interesting and correct questions that they were unable to solve, due to incomplete or incorrect reasoning. This observation led us to the concept of *AI-assisted creation of evaluation datasets*. Our process may also be of interest for human pedagogy since it begins with the extraction of core "skills" from existing math datasets, which serve as the foundational elements of mathematical questions. The current paper focuses on the MATH dataset (Hendrycks et al., 2021), a mainstay of LLM evaluation in recent years.

Starting with a list of mathematical skills extracted from the MATH dataset using recently discovered methods (Didolkar et al., 2024), we focused on creating questions that involve one skill from pre-algebra and algebra portions of the MATH dataset and one other skill randomly sampled from different sections of MATH. Our generation pipeline uses carefully crafted prompts and multi-turn interactions with leading models to significantly improve the generation of high-quality questions and candidate answers.

In our AI-assisted process, human experts played a crucial role. Using the (question, answer) pairs generated by LLMs and leveraging API access to leading models, experts identified promising questions—-often those incorrectly answered by the LLMs but containing many correct ideas. Experts then refined these questions to enhance their engagement value and provided gold-standard answers. The AI-assisted process not only boosted human productivity but also resulted in high-quality, novel questions distinct from those in existing datasets.

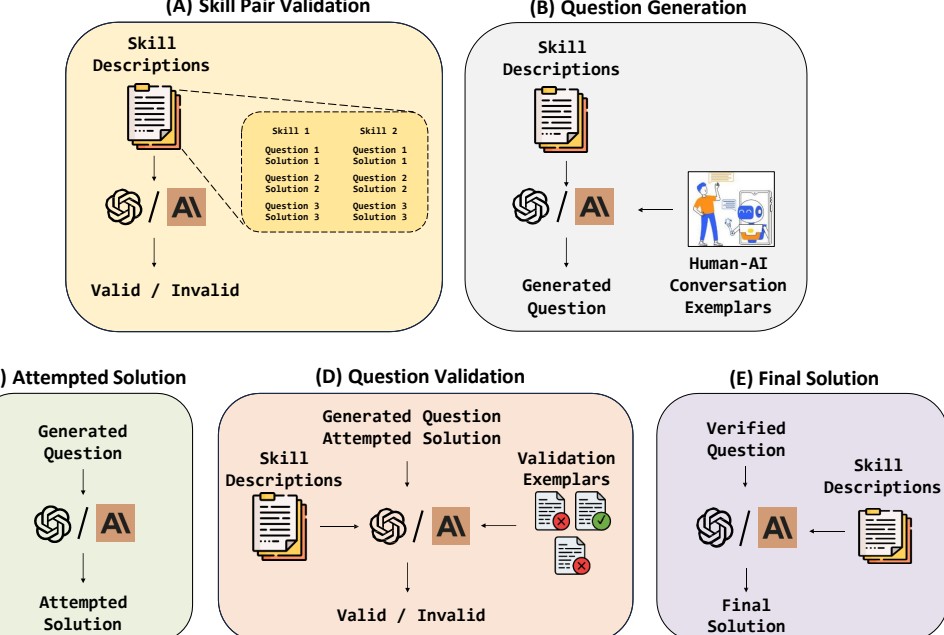

Figure 1: **AI-assisted question generation**: This figure illustrates a five-step AI-assisted pipeline for generating high-quality questions. **(a) Skill Pair Validation** - The model first validates that the given pair of skills are distinct and not too similar. **(b) Question Generation** - Using the validated skill pair, the model generates a question that necessitates the application of both skills for its solution. **(c) Attempted Solution** - Given the generated question, the model is asked to attempt a solution to the question while taking a *defeatist* approach. **(d) Question Validation** - The model validates the generated question based on the attempted solution, checking for correctness, skill rigor, clarity, and other quality criteria. **(e) Final Solution** - Valid questions are re-solved by the model using advanced techniques like in-context prompting and majority voting to enhance the accuracy of the final solution.

**Importance of using random pairs of skills:** We think that this is key driver of improved diversity and difficulty among generated questions. Recall that MATH dataset is neatly partitioned into sub-areas such as "Geometry" and "Number theory." Requiring generated questions to combine skills from two subareas (e.g., a question linking area-and-perimeter calculations with prime number knowledge) necessitates "out of distribution" thinking; some examples appear in Section 4.1. Such questions challenged all LLMs as well as our human raters. Our new dataset is much harder than MATH for all models. For instance, if a model has a success rate $0.5$ on MATH, then success rate on our new dataset is closer to $0.5^2 = 0.25$. This trend is more general, and Section 3.2 suggests that this is an indication that the average question of MATH$^2$ indeed requires nontrivial use of two distinct underlying skills, which is the reason for naming this new dataset as MATH$^2$. We believe that our methodology could also introduce fresh perspectives into math instruction for AI and human learners.

**Connection to Scalable Oversight:** This notion (Bowman & etal, 2022) looks ahead to how humans might supervise and check AI systems that potentially outperform humans in many relevant skills. While typically discussed in the context of alignment and safety, the concept is pertinent here. How can human experts reliably evaluate LLMs' understanding of high-school or freshman-level math when these models have already been trained on all available exams and textbooks? Could human-AI collaboration lead to more novel evaluations?

**Paper organization:** Sections 2 describes our design methodology and generation pipeline for MATH$^2$. Section 3 discusses the performance of many open-source and proprietary models on MATH(Hendrycks et al., 2021) as well as on the new MATH$^2$ dataset of 210 questions (see Table 2 ). Section 3.2 discusses the interesting relationship between MATH and MATH$^2$ scores. Section 3.3 shows that MATH$^2$ questions are more useful than MATH questions when used as in-context exemplars for various LLMs. Section 4 sheds some light on interesting behaviors and failure modes of leading LLMs that we observed during the question generation process.

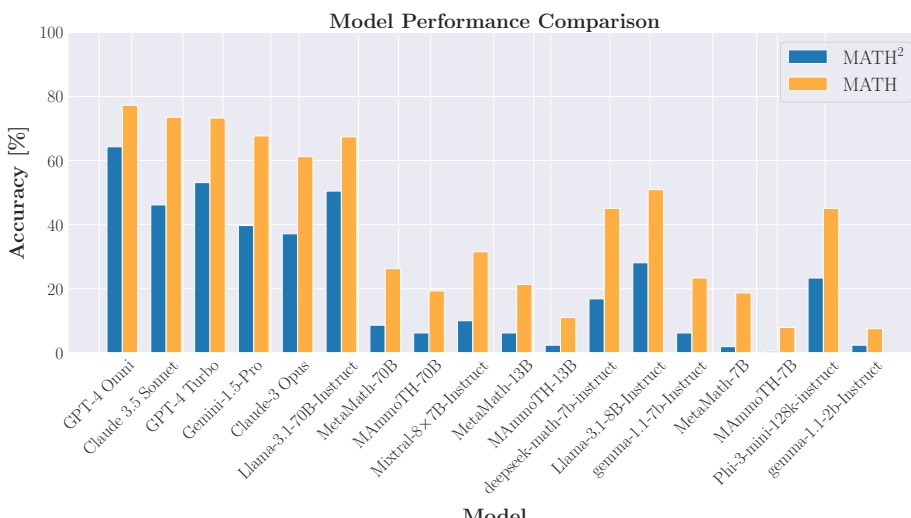

Figure 2: **Comparison of Zero-Shot Performance of Various Models on MATH and new Dataset MATH².** - This figure illustrates the zero-shot Chain of Thought (CoT) performance of both open-source and proprietary models on two different datasets: MATH and our generated dataset. Across the board, models demonstrate a lower performance on the generated dataset compared to MATH. Proprietary models exhibit the smallest decrease in performance, while smaller models within the same family experience more significant performance drops. Detailed numerical values related to this comparison are available in Table 2.

## 2  PIPELINE FOR AI-ASSISTED QUESTION GENERATION

We present a structured approach to generating challenging mathematics questions by combining the capabilities of large language models (LLMs) and human expertise. Given below is a high-level overview of the process before delving into the details of each step.

We begin our pipeline with **skill extraction** - identifying and cataloging distinct mathematical skills from a dataset, as described in Didolkar et al. (2024). This step creates a repository of skills linked to specific questions. The motivation behind this is to systematically generate and analyze questions that require specific skills, ensuring a comprehensive evaluation framework.

Next, we focus on **generating questions that combine pairs of distinct skills** to increase their difficulty. By using advanced models like GPT-4, Claude and Gemini, and incorporating in-context examples of multi-way interactions between AI and humans, we enhance the models' performance in generating complex questions. This step aims to produce challenging questions that robustly assess problem-solving abilities.

The final step involves **screening and validation** to filter out invalid or flawed questions. This rigorous process includes evaluating and solving the questions to identify hidden flaws, such as computational intractability or logical inconsistencies. Advanced techniques like in-context exemplars and self-consistency further ensure the accuracy and quality of the solutions. This step is crucial for maintaining the integrity and reliability of the generated questions and their solutions. Overall, each step in the pipeline is designed to systematically enhance the quality and difficulty of questions, providing a robust and comprehensive evaluation of mathematical skills.

Motivated by these challenges, we employ a five-step approach to generate difficult math questions using advanced models. For each round of generation, we randomly sample a pair of skills and three sample question-solution pairs corresponding to each skill from the skill repository. These reference examples are sourced from the MATH dataset.

**Step 1: Skill Pair Validation.** We begin by asking the LLM (GPT-4 or Claude) to validate the skill pair by assessing the qualitative similarity of the two skills. Reference examples are provided in-context to enrich the model's understanding of the skills. If the model deems the skills too similar,

they are flagged and excluded from question generation, as similar skills might lead to simpler questions.

**Step 2: Question Generation.** Next, we prompt the LLM to generate a question and a brief solution requiring the application of both skills in the sampled pair. We specify two conditions to ensure high-quality questions: the question should either require an exact answer or specify that an approximate answer is acceptable, and it should ask for only a single final result. In-context, we provide two multi-turn conversations between a human and an AI assistant. These conversations demonstrate the human providing feedback on the AI-generated questions, which the AI then refines. This helps the model anticipate and avoid practical issues, such as insufficient involvement of skills or logical inconsistencies. Appendix A.6 provides examples of the responses of different models in the question generation step.

**Step 3: Solution Attempt.** The model then attempts a solution to the generated question, adopting an adversarial approach to identify flaws such as insufficient information, ambiguity, self-contradiction, or excessive computation. If any issues are found, the model stops solving and clearly states the problems. Otherwise, it completes the solution. During this step, the model does not receive the skill names or reference examples to ensure unbiased problem-solving.

**Step 4: Question Validation.** We give LLM the generated question and its solution for validation against a fixed rubric consisting of seven criteria:

- Single Answer Requirement: The question should ask for only one final answer.

- Exact Answer Requirement: There should be only one exact answer, unless approximations are explicitly stated.

- Dual Answer Requirement: The question must necessarily and sufficiently involve the application of both skills, with difficulty comparable to or greater than the reference examples.

- Clarity and Completeness: The question should be clear and contain all necessary information.

- Computational Tractability: The question should not require overly complex computations.

- Realism and Logic: The scenario should be realistic and logically consistent.

- Syntax and Grammar: The question should be grammatically correct and clearly written.

The model uses reference examples and validation exemplars - model generated examples of validating questions, to facilitate this step. We employ majority voting (maj @ 4) to enhance robustness.

**Step 5: Final Solution and Re-validation.** For questions classified as valid, we ask the LLM to re-solve the question to obtain a final solution. Reference examples are provided in-context to improve the model's understanding. We use majority voting (maj @ 4) to ensure consistency. If all the answers obtained in this step are unique, indicating potential ambiguity, the question is discarded.

The questions obtained from the above pipeline are further screened by humans. This structured approach not only generates challenging and novel math questions but also ensures their quality through rigorous validation, effectively combining the strengths of AI and human oversight. For detailed examples of prompts used at each step, refer to Appendix A.7.

## 3 EXPERIMENTS AND FINDINGS

Through our experiments, we demonstrate the difficulty and quality of the MATH$^2$ while also analyzing the behavior of different models on this task of *compositional generalization*. Firstly, we evaluate a wide range of models spanning a large range of parameter counts on MATH$^2$ and compare against their performance on MATH (Hendrycks et al., 2021) which is the base dataset used for extracting skills, showing that the MATH$^2$ is necessarily harder than MATH. Next, we further demonstrate the difficulty and quality of questions in MATH$^2$ by showing that they are better in-context exemplars as compared to standardly used exemplars. We describe the experimental setup below.

Table 1: **Human Verification Statistics:** Out of a total of 210 examples in MATH$^2$, 139 (66.19%) were such that either the question or the solution generated by the model were modified by the annotator before being included in the final dataset. These modifications were made in order to increase the difficulty of the questions or correct the questions or solutions.

| # of Modified Questions ($A$) | # of Modified Solutions ($B$) | # of $A \cup B$ | Dataset Size |
|:---:|:---:|:---:|:---:|
| 64 | 136 | 139 | 210 |

### 3.1 EXPERIMENTAL SETUP

We follow the pipeline proposed in (Didolkar et al., 2024) to extract skills from the MATH dataset (Hendrycks et al., 2021). The MATH dataset encompasses seven high-level topics, allowing us to identify and extract finer-grained skills within each topic and label each question accordingly. At the end of the skill-extraction process, we identify a set of 114 skills. We then remove a few simple skills, such as `basic_arithmetic` and `arithmetic_operations`, before using the remaining set to generate questions using the proposed approach. We generate and verify 210 difficult questions to create the MATH$^2$ dataset. Out of the 210 questions, 116 questions were generated using GPT-4 Turbo, 3 using GPT-4 Omni, 51 using Claude-3 Opus and 40 using Gemini-1.5-Pro. Figure 5 shows the distribution of skills in MATH$^2$.

Table 1 presents details of the changes made to the questions during the human verification process. Out of 210 question-solution pairs included in MATH$^2$, 139 underwent some form of modification by the human annotators before being included in the dataset. Out of the 64 questions modified, 3 were minor modifications to improve the clarity of the question. Another 22 modifications were minor modifications, which nevertheless affected the meaning of the question and changed the final answer. But 39 modifications were significant; either making the given questions harder, or correcting them, or making them more interesting (i.e., less tedious) for humans.

Table 2: **Comparison of Zero-Shot CoT Performance (Accuracy) on the Generated Dataset vs. MATH Test Set**: GPT-4 Omni demonstrates the least drop in percentage terms (16.73%) whereas MAmmoTH-7B shows the highest relative drop (93.92%).

| Model | MATH$^2$ (Y) | MATH (X) | % Drop |
|:---:|:---:|:---:|:---:|
| GPT-4 Omni | 64.29% | 77.21% | **16.73%** |
| Claude 3.5 Sonnet | 46.15% | 73.54% | 37.24% |
| GPT-4 Turbo | 53.11% | 73.27% | 27.51% |
| Gemini-1.5-Pro | 39.71% | 67.70% | 41.34% |
| Claude 3 Opus | 37.14% | 61.20 % | 39.31% |
| Llama-3.1-70B-Instruct | 50.48% | 67.40% | 25.10% |
| MetaMath-70B | 8.61% | 26.27% | 67.22% |
| MAmmoTH-70B | 6.19% | 19.31% | 67.94% |
| Mixtral-8×7B-Instruct | 10.00% | 31.52% | 68.27% |
| MetaMath-13B | 6.19% | 21.32% | 70.96% |
| MAmmoTH-13B | 2.38% | 10.99% | 78.34% |
| Deepseek-math-7b-instruct | 16.83% | 45.05% | 62.64% |
| Llama-3.1-8B-Instruct | 28.09% | 50.92% | 44.83% |
| Gemma-1.1-7B-Instruct | 6.19% | 23.36% | 73.50% |
| MetaMath-7B | 1.91% | 18.69% | 89.78% |
| MAmmoTH-7B | 0.48% | 7.90% | **93.92%** |
| Phi-3-mini-128k-instruct | 23.34% | 48.29% | 51.67% |
| Gemma-1.1-2B-Instruct | 2.38% | 7.52% | 68.35% |

As for the solutions, 136 out of the 210 solutions originally generated by the model were modified to correct them or improve their clarity. This includes solutions which had to be modified because of modifications in the corresponding question.

In total, 33.81% of the question-answer pairs in MATH$^2$ appear exactly as phrased by their LLM creators.

We evaluate the generated set of questions on a variety of language models, both small and large. Specifically, we assess the MetaMath (Yu et al., 2023), MAmmoTH (Yue et al., 2023), Gemmma (Team et al., 2024b), and Llama-3.1 series (Dubey et al., 2024), Phi-3 (Abdin et al., 2024), deepseek-math (Shao et al., 2024) as well as one Mixture-of-Experts model Mixtral-8×7B-Instruct (Jiang et al.,

2024). Additionally, we include evaluations of larger proprietary models such as GPT-4o, GPT-4 Turbo[1] (OpenAI, 2023), Gemini-1.5-Pro (Team et al., 2024a), Claude 3.5 Sonnet [2] (Anthropic, 2024) and Claude-3 Opus[3]. We compare the performances of these models on our generated questions against their performance on the MATH dataset (Hendrycks et al., 2021). We further report several ablation studies on MATH$^2$ in Appendix A.3.

For generating responses, we use the MAmmoTH (Yue et al., 2023) evaluation suite. The responses are graded using a GPT-4 grader, where GPT-4 Omni checks the correctness of a solution response against the ground truth solution. During response generation, we set the temperature to 0 and top_p to 1 for all models. All necessary compute details are discussed in Appendix A.3

## 3.2 Performance across the two datasets: A surprising pattern

Table 2 shows that all tested models have significantly lower performance on MATH$^2$ than on the original MATH dataset. Denoting $Y$ as the performance on MATH$^2$ and $X$ as the performance on MATH, the percentage drop $100(X - Y)/X$ for frontier models — GPT-4 Omni, GPT-4 Turbo, Gemini-1.5-Pro, Claude-3.5-Sonnet and Claude 3 Opus — ranges from 16.73% to 41.34%. MAmmoTH-7B, a specialist math model, shows the largest drop at **93.92%**.

The fact that performance drops for all models should *not* be too surprising, since as noted, the MATH$^2$ questions, by combining skills from different subareas of MATH, could be seen as "out of distribution (OOD)." This makes it tempting to interpret the percentage drop as a measure of a model's (lack of) "OOD-resilience." For instance, very large percentage drops seen with open-source models MetaMath and MAmmoTH feel understandable since their training used synthetic data generated using seed questions from MATH and GSM-8k. Lack of diversity in such

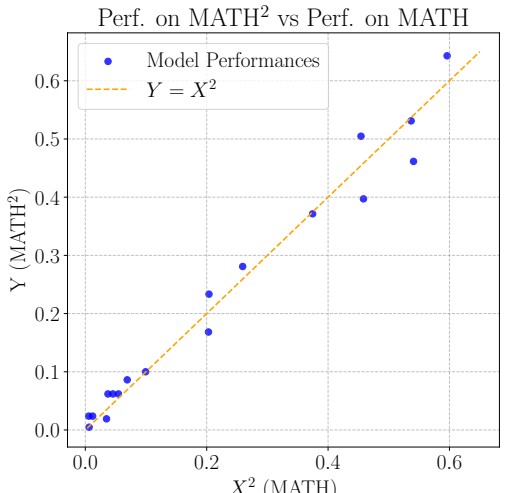

Figure 3: Relation between the performance of models on MATH$^2$ ($Y$) vs the square of their performances on MATH ($X^2$). As can be seen from the plot, $Y \approx X^2$. See Appendix A.3 for the best-fit quadratic curve, which is slightly different.

synthetic data is known to cause overfitting to the dataset being imitated. Similarly, GPT-4o and Claude Sonnet 3.5 are suspected to also have been extensively trained with synthetic data. Although their MATH performance is similar, Sonnet 3.5 has worse MATH$^2$ performance, which might suggest lower quality/diversity in its synthetic data.

However, in our opinion, the overall pattern among proprietary models of similar size does fit with the OOD story. A much simpler explanation pops out when we plot $Y$ vs $X^2$ (Figure 3 and Figure 4(a)): we find a linear relationship $Y \approx X^2$! This implies that the relative drop in performance of the models is well-predictable from just their performance on MATH, and does not require taking their training details into account!

Why should the two scores be expected to have this relationship? Here is a natural (albeit heuristic) explanation. Suppose there are $N$ skills and $s_i$ denotes the success rate of the model at correctly applying the $i$th skill. Then, its $X$ value should reflect the average[4] of the $s_i$'s. Furthermore, on a random question using the $i$th and $j$th skill, the probability that the model correctly answers it should be $s_i s_j$, since it has to successfully apply both skills. If the questions are created using pairs of skills chosen randomly and independently, then the $Y$ value will be the average value of $s_i s_j$'s, which by independence will be roughly $X^2$.

---

[1]Points to `gpt-4-turbo-2024-04-09` at the time of writing

[2]Points to `claude-3-5-sonnet-20240620` at the time of writing

[3]Points to `claude-3-opus-20240229` (Anthropic, 2024) at the time of writing

[4]With perhaps a small correction factor if the skills are not evenly distributed among the questions

Table 3: Performance of models on MATH under two different prompting strategies. **MAmmoTH 4-shot CoT** prompting involves 4-shot prompting with exemplars taken from the MAmmoTH (Yue et al., 2023) evaluation suite. **Skill Based 4-shot CoT** (Didolkar et al., 2024) consists of using 4 exemplars which are retrieved from the training set of MATH based on which skill is required to solve the given question (as determined by GPT-4). **Proposed 4-shot CoT** prompting consists of 4-shot prompting with exemplars taken from MATH$^2$. These exemplars are retrieved such that one of the two skills in each exemplar is the same as the skill required by the question at hand, as labeled by GPT-4. In **Proposed + Skill Based 4-shot CoT** we supplement the exemplars retrieved from MATH$^2$ with exemplars from MATH training set, for skills that are present in < 4 questions in MATH$^2$. We show that few-shot prompting with exemplars retrieved from the generated set of questions (MATH$^2$) consistently outperforms vanilla few-shot prompting with relative gains of upto 12.79% over the baseline (for for Llama-3.1-70B-Instruct (Dubey et al., 2024)).

| Method | GPT-4O | GPT-4T | Llama-3.1-70B-Instruct | MetaMath-70B | MAmmoTH-70B | Mixtral-8×7B-Instruct |
|---|---|---|---|---|---|---|
| MAmmoTH 4-shot CoT | 76.67% | 71.89% | 58.15% | 25.77% | 18.45% | 30.77% |
| Skill Based 4-shot CoT | 78.32% | 72.77% | 57.81% | 25.42% | 18.20% | 30.31% |
| Proposed 4-shot CoT | **78.74%** | **72.96%** | **65.59%** | **27.54%** | **21.23%** | **34.23%** |
| Proposed + Skill Based 4-shot CoT | 78.53% | 71.51% | 61.95% | **27.54%** | 20.41% | 33.95% |

This reasoning in fact suggests that our pipeline has created questions that genuinely required applying two very distinct skills (as opposed to, say, requiring primarily skill $i$, and mildly using skill $j$). The discovered relationship suggests further that if we could create questions where each combines $k$ skills, we might see the relationship $Y \approx X^k$, which would tend to further magnify performance differences between models.

## 3.3 GENERATED QUESTIONS ARE EFFECTIVE IN-CONTEXT EXEMPLARS FOR MATH.

A possible test for the quality of a Q&A pair on similar topics as MATH dataset is whether performance on MATH improves when using these as in-context exemplars.

We test as follows. Recall that MATH has 7 sections. Exemplars for a section are chosen from the section area. However, by design, our new questions cross section boundaries. Furthermore, they are higher quality than MATH questions. We implemented a new procedure to retrieve in-context exemplars from MATH$^2$ based on the skill requirements of the current question.

Since MATH$^2$ is limited in size, it does not cover all the skills extracted during the skill extraction process, containing 93 out of 114 skills. Figure 5 shows the distribution of different skills in the dataset. We filtered the MATH test set to remove examples requiring skills not present in the generated dataset, resulting in the removal of 913 test examples. During evaluation on the filtered MATH test set, for each question $Q$ labeled with skill $a$ ($a \in \mathcal{S}$, where $\mathcal{S}$ is the set of extracted skills), we retrieved in-context exemplars from the MATH$^2$, ensuring each exemplar involved skill $a$. We used four such exemplars per question (i.e., 4-shot CoT (Wei et al., 2022)). To handle skills represented by fewer than four examples in MATH$^2$, we run two experiments: (A) **Proposed 4-shot CoT**: If a given skill is represented by $n$ examples in the MATH$^2$, where $n < 4$, we use $n$ in-context examples instead of 4 exemplars. (B) **Proposed + Skill Based 4-shot CoT**: If a given skill is represented by $n$ examples in MATH$^2$, where $n < 4$, we supplement $4 - n$ exemplars for that skill from MATH training set. The relevant in-context exemplars in MATH training set are determined by following the methodology proposed in Didolkar et al. (2024). We compared the performance of models using these targeted prompting strategies against two baselines: (C) **MAmmoTH 4-shot CoT**: The 4 in-context exemplars are taken from the MAmmoTH evaluation suite (Yue et al., 2023). (D) **Skill Based 4-shot CoT**: We use skill-based prompting as proposed in Didolkar et al. (2024), where the in-context exemplars are selected from the MATH training set, in accordance to the skill required by the question at hand, as determined by GPT-4.

Table 3 presents the results of this comparison. The two prompting strategies using questions from MATH$^2$ as in-context exemplars, clearly outperform the two baselines. We conclude that the MATH$^2$ questions, due to their difficulty and skill relevance, serve as effective in-context exemplars. Performance gains would likely be more significant with larger datasets generated using our approach, reducing the need to supplement with external exemplars.

## 4 OBSERVATIONS FROM THE QUESTION GENERATION PROCESS

The question generation pipeline described in Section 2 was developed through an iterative process of refining prompts and design choices, and evaluating their impact on the quality of the final questions and solutions. Notably, the inclusion of the *attempted solution* and *question validation* steps significantly enhanced the pipeline's effectiveness. Despite the sophistication of the pipeline and prompts, we still observe instances where models fail to follow the given instructions. This section highlights prominent failure modes at various stages of the pipeline, which human raters need to be aware of. Additionally, we explore some intriguing behaviors of the models where they successfully create interesting and creative questions. Section 4.1 details the role of human raters in improving these questions.

### 4.1 CREATIVE QUESTIONS: EXAMPLES OF SYNERGY FROM HUMAN-AI INTERACTION

The models frequently produced interesting and creative questions, although they often failed to generate correct solutions. In these cases, the incorrect solutions usually contained enough correct ideas for a human to quickly complete them.

Human annotators were tasked with verifying the validity of the questions and the correctness of the solutions. They were instructed to look out for any failure modes discussed in Appendix A.1. Their responsibilities included ensuring that the created questions actually employed the intended math skills, and improving the questions in terms of readability, quality, and difficulty when possible. They were encouraged to suggest changes that would make the problems harder for automated tools to solve while allowing easier or more elegant solutions for humans. The following examples illustrate this process:

> **Example:** *Original Question: Find the smallest positive integer $k$ such that $k^3 - k$ is divisible by both 9 and 10, and the sum of digits of $k$ in its decimal representation is a prime number.*

Our human team had not encountered such questions before. It requires recognizing that $k^3 - k = k(k-1)(k+1)$ is always divisible by 2 and 3. Thus, $k$ must be such that $k(k-1)(k+1)/6$ is divisible by 15 (both 3 and 5). Additionally, the sum of the digits of $k$ must be a prime number, and ensuring such conditions is challenging even for powerful LLMs.

> **Example:** *Original Question: Consider a collection of red, blue, and green beads arranged in an infinite series. The beads alternate in color, starting with red, then blue, then green, and this pattern repeats indefinitely. The number of beads in each colored section follows the pattern of powers of 2: the first red section has 2 beads, the first blue section has 4 beads, the first green section has 8 beads, the second red section has 16 beads, and so on. If a bracelet is made using a continuous, unbroken sequence of exactly 20 beads from this series, and each bead has a length of 0.5 units, how many different bracelets can be made such that the perimeter of the bracelet is an integer value?*

The original question combined elements in a novel way. The human rater modified the question to change the sequence size from 20 to 6 beads, maintaining the essential difficulty while making it more elegant for humans. All tested models failed on the modified question.

> **Example:** *Original Question: A container initially contains 500 mL of water. A scientist adds water to the container $\frac{1}{4}$ of the current amount every minute. After how many minutes will the container first contain more than 1 L but less than 2 L of water?*
>
> *Modified Question: A container starts with 500 mL of water. Each minute, the scientist adds water equal to $1/2$ of the current amount. What is the smallest positive integer $n$ such that the number of liters of water in the container is never in the interval $[n, n+1]$?*

This was one of many questions the models created about exponential growth and geometric series, possibly similar to standard math test questions. The human slightly altered it to simplify calculations

by hand and substituted a different condition that the models found challenging, while humans could easily estimate an approximate answer and then verify.

> **Example:** *Original Question: Consider the sequence defined recursively by $a_1 = 1$ and $a_{n+1} = 2a_n + n$ for all $n \geq 1$. What is the product of the first five terms of this sequence?*
>
> *Modified Question: A sequence $a_n$ is defined as follows: $a_1 = 2$ and $a_n = 2^{n-1} + a_{n-1} + n$. What is the $\lfloor \log_2 a_{500} \rfloor$?*

An LLM can solve the original question through simple computation. The modified question, however, requires understanding an underlying pattern.

> **Example:** *Original Question: Find the sum of the smallest prime divisor and the largest prime divisor of the number $N = 15^4 + 16^4$.*
>
> *Modified Question: Find the sum of the two smallest prime divisors of $23^{17} + 17^{17}$.*

Models tend to adopt a brute-force approach to the original question by calculating $15^4 + 16^4$. After rephrasing, the number $23^{17} + 17^{17}$ is too large for direct computation, requiring understanding of arithmetic modulo a prime.

These examples highlight the essential role of human oversight in refining and improving the questions generated by LLMs, ensuring they are challenging, creative, and suitable for advanced mathematical problem-solving.

Despite the sophistication of our pipeline, models frequently exhibit several failure modes such as *insufficient involvement of skills*, *insufficient information*, *unsolvable or computationally intractable questions*, *nonsensical questions* and *deceitful solutions*. For a more detailed discussion and examples of questions in the various categories listed above, refer to Appendix A.1.

## 5  CONCLUSIONS

We introduced a framework that leverages the complementary strengths of humans and AI to generate new, challenging mathematics questions. Building on recent insights into LLM metaknowledge, we use LLMs to extract and name key skills necessary for solving math problems. Using these insights, we developed a pipeline that employs named skills from the well-known MATH dataset, and leverages multi-turn interactions with advanced LLMs to generate questions that combine pairs of skills. These questions were subsequently reviewed and refined by human raters. The proposed pipeline produced questions with greater novelty and difficulty compared to those in the original MATH dataset. The resulting new math evaluation – **MATH**$^2$, assesses the same skills as the MATH dataset but is significantly more challenging for leading models because each question involves two skills from different parts of MATH. GPT-4-T and GPT-4-O exhibited the smallest performance drops, which aligns with the subjective evaluations of our human raters. Additionally, we demonstrated that providing the newly generated questions as in-context examples improved GPT-4-O's performance on the MATH dataset more effectively than examples sourced directly from the MATH dataset. This finding further validates the quality of the questions produced by the proposed pipeline. We plan to release detailed information about our pipeline to encourage further research and development in the field of open-source math models.

**Limitations and Future Work.** Our pipeline incurs moderately high costs due to extensive API-based use of frontier models as well as significant human verification. To improve efficiency, future work should focus on using open weights models and optimizing prompting strategies to produce higher-quality questions initially, thereby reducing the need for extensive filtering. Additionally, reducing human verification through the development of automated validation tools is crucial. This could include leveraging code generation and autoformalization capabilities of LLMs to generate responses which can be compiled using compilers or interpreters. Enhancing our pipeline by integrating a training-based feedback loop, where the model is trained on the questions that pass human verification, could further streamline the process by progressively improving question quality. These measures will reduce dependency on expensive proprietary models, lower overall operational costs, and maintain or even enhance the quality of the generated math evaluation benchmarks. Looking ahead, an even more exciting prospect is the potential application of the proposed framework to efficiently produce high-quality data in domains beyond mathematics.

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

## A    APPENDIX

Here we further analyze the quirks of the question generation pipeline and provide additional experimental details and results. In Appendix A.1, we discuss several failure modes of the models that we notice during the question generation process as well as interesting behaviors exhibited by the models throughout the pipeline and interesting creative questions that the models came up with. Appendix A.2 discusses the different considerations that human annotators were instructed to take into account while annotating and verifying the questions generated by the proposed pipeline. Appendix A.3 provides details about the compute used for running our experiments as well as some ablation studies on the MATH$^2$ dataset. In Appendix A.6 we provide examples of questions generated by different models in the **Question Generation** step (Section 2). Appendix A.7 gives a detailed description of the prompts used for each step in the question generation pipeline as well as evaluation of the models. It also provides a link to the skill exemplar repository and in-context exemplars used in the question generation process.

### A.1    FAILURE MODES AND INTERESTING BEHAVIORS

Despite the sophistication of our pipeline, models frequently exhibit several failure modes: (a) *Insufficient Involvement of Skills*: Models often generate questions that either miss one of the skills completely or require a very shallow application of one or both skills. For example, a geometry question may fail to involve ratio and proportion adequately, (b) *Insufficient Information*: Questions may lack essential details needed for solving, making them incomplete or ambiguous. For instance, a trigonometry question might omit necessary angles or distances, (c) *Unsolvable or Computationally Intractable Questions*: Some questions generated are either unsolvable or require excessive brute-force calculations, which are impractical for evaluating reasoning abilities, (d) *Nonsensical Questions*: Models sometimes produce questions that are logically inconsistent, confusing, or ambiguous, such as a probability problem with unclear parameters or an impossible geometry scenario, (e) *Deceitful Solutions*: Occasionally, models fabricate solutions to nonsensical or unsolvable questions, presenting incorrect logic as plausible reasoning and (f) *Finding a Needle in the Haystack*: Long and complex validation prompts sometimes cause models to confuse or overlook the specified skills, leading to incorrect evaluations.

**Insufficient involvement of skills.**    Despite clearly specifying that solving the question should necessarily require a rigorous application of both skills, the models often generate questions that either miss one of the skills completely or require a very shallow application of one (while the other one is sufficiently involved) or both skills. This is the most prominent failure mode of the models in the context of question generation. This leads to potentially easy questions, defeating the purpose of skill composition. Consider the question given below which was generated by Claude Opus when

asked to combine the skills `ratio_and_proportion` and `geometry`.

> **Example:** `Question: A square garden is to be divided`
> `into 4 smaller square plots by two paths that are 1`
> `meter wide and cross each other at right angles. The`
> `paths run North-South and East-West, splitting the`
> `garden symmetrically. If the total area occupied by`
> `the paths is 36 square meters, find the side length of`
> `the original square garden.`

Upon careful examination of the question, we note that although the question tests `geometry`, the involvement of `ratio_and_proportions` is practically non-existent. Further, the question validation step in some cases also fails to identify these flaws. Supplying multi-turn human-AI interactions where the user prompts a chatbot to generate a question combining two skills, in-context during the generation step helps the models to avoid such questions to a certain extent. Further, to make the question validation step more robust to such questions, we prompt the model to ensure that the complexity of each skill application in the question being validated in similar to or more than the complexity of these skills in the reference examples present in the skill descriptions. The combination of these two techniques helps us nearly eliminate questions where the absent one of the skills is absent completely and reduce questions involving shallow application of skills to a significant extent.

**Insufficient information in the questions.** Another common failure mode of the pipeline is the generated questions missing information or details essential for solving the question. For example in the question given below which is supposed to combine the skills `understanding_and_applying_floor_and_ceiling_functions` and `basic_trigonometry`, lacks sufficient detail about the inclinations and elevations of the paths relative to the streetlight's position which is necessary to answer the question.

> **Example:** `Question: Consider a scenario where you need`
> `to install a new streetlight at a point such that it`
> `illuminates two paths meeting at a point, each path`
> `making an angle of 45° with the horizontal. The light`
> `from the streetlight reaches a maximum distance of`
> `10 meters on flat ground. You are to install the`
> `streetlight at the height of` $h$ `meters (where` $h$ `is`
> `the ceiling of the maximum distance the light reaches`
> `horizontally) such that the edge of the light's reach`
> `just touches the ground at the end of each path.`
> `Determine the height` $h$ `at which the streetlight should`
> `be installed.`

To screen such questions, we include and explicit clause in the question validation prompt as described in Section 2. Moreover, we also notice that the inclusion of the *solution attempt* step improves the chances of detecting such errors since the missing information may not always be apparent from just the question itself. In such cases, attempting a solution (with a defeatist approach) can help detect such flaws.

**Unsolvable or Computationally Intractable Questions.** There are instances when the model generates questions which are unsolvable. For example the question given below has no solution which satisfies all three constraints (i.e., the area of the rectangle being 360 and the sides belonging to the two arithmetic progressions defined in the question.)

> **Example:** `Question 1: There's a rectangle with an area`
> `of 360 square units. The length of the rectangle is`
> `part of an arithmetic sequence starting at 5 and with`
> `a common difference of 7. If the other side of the`
> `rectangle is also part of an arithmetic sequence with`
> `the first term 10 and common difference 3, find the`
> `length of the shortest side of the rectangle.`

In other instances, the model generates questions that are computationally intractable or require manually and tediously iterating through a long sequence of values. For example, solving the question given below requires manually calculating the first 100 terms of the sequence to find the sum

> **Example:** `Question 2: Consider an infinite series of`
> `numbers arranged in sections, where the` $n$`th section`
> `contains the first` $\binom{n+1}{2}$ `positive integers that are`
> `divisible by` $n$ `but not by any smaller positive integer`
> `(except 1). For example, the 1st section contains 1,`
> `the 2nd section starts with 2, 4, 6, 10, 12, and 16`
> `the 3rd section starts with 3, 9, 15, 21, 33, ... and`
> `so on. Let` $S$ `be the sum of the first 100 terms of`
> `this series. Find the sum of the digits of` $S$`.`

While technically not wrong, such questions are not ideal for evaluating the *reasoning* abilities of the models since they mostly involve brute force calculations. Further, in cases where the sequence of calculations is very long, the LLM's performance may be bottlenecked by other limitations such as the context length of the model.

Thus, we strive to filter such questions out. We add an explicit condition to check for computational tractability and solvability of the generated questions in the verification prompt. This check is assisted by the *solution attempt* produced by the model which will potentially point out any such problems.

**Nonsensical Questions.** In several cases, the model comes up with questions which are nonsensical - confusing, incomprehensible, logically inconsistent or ambiguous. Consider the question given below.

Given below is an example of a question which is logically inconsistent. More concretely, a square plot of land whose side length is equal to the radius cannot fit inside the quarter-circle.

> **Example:** `Question: A garden is designed in the shape`
> `of a quarter-circle with a radius of 8 meters. A`
> `square plot of land with a side length equal to the`
> `radius of the quarter-circle is placed inside this`
> `garden such that two of its sides are along the`
> `straight edges of the quarter-circle boundary. If`
> `the square plot of land is to be tiled entirely with`
> `square tiles each of area 64 square centimeters, what`
> `is the total number of tiles required?`

We add checks for such cases in the question validation prompt. Further, at the end of the final solution step (maj @ 4), we further check for cases where the final answer produced in all the 4 self-consistency trials are unique. If all answers are unique, we discard the question. The rationale behind this being that it is highly likely that the model produces a different answer every time due to some inherent ambiguity in the question which was not detected in the *solution attempt* and the *question validation* checks.

**Deceitful Solutions.** Although rare, we encounter cases where the model makes up solutions even though the question is nonsensical or cannot be solved with the amount of information provided in the question. This happens very commonly in the solutions which are generated in the *question generation* prompt. Thus, we do not use these solutions and include the *final solution* step where the model is asked to solve the question again. Although most of such solutions and thus questions are screened out in the *question validation* step and consistency check at the end of the *final solution* check, in rare cases we see this behavior in the solution produced after the *final solution* step as well. Given below is one such example.

> **Example:** `Question: Consider the trigonometric`
> `identity` $\sin^2(x) + \cos^2(x) = 1$ `and the polynomial` $P(x) =$
> $x^4 - x^2 - 12$`. Using` $x = \sin(\theta)$`, solve` $P(x) = 0$ `for` $\theta$ `in the`
> `interval` $[0, 2\pi)$`.`

While solving this question, the model arrives at the conditions $sin(\theta) = 2$ or $sin(\theta) = -2$. Clearly, these conditions have no solutions since $-1 \leq sin(\theta) \leq 1, \forall \theta \in [0, 2\pi]$. However, the model goes on to argue that the the closest values to 2 and -2 in the range of $sin(\theta)$ are 1 and -1, and thus, goes on to solve the question for $sin(\theta) = 1$ or $sin(\theta) = -1$.

**Finding a needle in the haystack.** In an attempt to make the *question validation* step robust to as many failure modes as possible, we come up with a long and sophisticated prompt (see Appendix A.7.4 for an example of this prompt). To elaborate, the validation prompt comprises of 1.) skill descriptions consisting of 3 exemplars for each of the 2 skills, 2.) 6 very long validation exemplars 3.) rest of the prompt consisting of the 7 conditions described in Section 2. In prior iterations of the validation prompt, the names of the two skills which are supposed to be combined in the question, were mentioned only once throughout the prompt. In such a case, we observed that more often than not, model confused the pair of skills at hand, with skills mentioned in the validation exemplars. We also observed cases where the model was unable to locate the original skill names, came up with its own skill names depending on the skill exemplars provided in the skill descriptions. More specifically, while checking the question for **Dual Skill Requirement** in Section 2, the models would check the question for the presence of skills distinct from the pair of skills at hand. We attribute this phenomenon to the the *lost in the middle* or *finding a needle in the haystack* challenges faced by LLMs. We provide an example of the phenomenon below. Notice how in Point 3 of the reasoning trace the model checks the question for the presence of two completely unrelated skills. We were able to nearly eliminate this problem by replacing referential terms (such as *above given skills*, *these skills*, etc.) to the two skills by the skill names.

> **Example:** Skill Pair to be checked for:
> [understanding_and_manipulation_of_rational_
> functions, division_and_remainders]
> Question: A fair 6-sided die is rolled 4 times. What
> is the probability that the product of the numbers
> rolled is divisible by 36? Express your answer as a
> simplified fraction.
> Reasoning trace of the model:
> # REASONING
> Let's evaluate the given question and solution against
> the specified criteria:
> 1. **Single Answer Requirement**: The question asks
> for a single value - the probability of a specific
> event occurring. This satisfies the single answer
> requirement.
> 2. **Exact Answer Requirement**: The solution
> provides an exact answer in the form of a simplified
> fraction (61/2386). No approximations or assumptions
> are made, fulfilling the exact answer requirement.
> 3. **Dual Skill Requirement**: The question requires
> knowledge and application of two distinct mathematical
> skills:      - Probability theory: Calculating the
> probability of specific outcomes in a series of die
> rolls.   - Number theory: Understanding divisibility
> rules for 4 and 9, and how they apply to the product
> of numbers.
> The solution demonstrates a rigorous application of
> both skills, using probability calculations (such as
> the binomial probability for rolling a 4 at least
> once) and number theoretic concepts (divisibility
> rules for 4 and 9) to arrive at the final answer. The
> complexity of skill application is on par with the
> provided examples.
> 4. **Clarity and Completeness**: The question is
> clearly stated and provides all necessary information

```
(fair 6-sided die, rolled 4 times, product divisible
by 36) to solve the problem.  No ambiguity or missing
information is present.
5.  **Relevancy of Information**:  All given
information is relevant and used in the solution.
There are no extra details that do not contribute to
solving the problem.
6.  **Realism and Logic**:  The scenario of rolling
a die multiple times and calculating probabilities
is a realistic and commonly encountered problem in
probability theory.  The data (6-sided die, 4 rolls)
is logically consistent.
7.  **Syntax and Grammar**:  The question is
grammatically correct and clearly written, with no
room for misinterpretation.
The solution is well-structured and easy to follow,
breaking down the problem into clear steps and
explaining the reasoning behind each step.  The
mathematical reasoning is sound, correctly applying
probability and number theory concepts to arrive at
the final answer.
# FINAL ANSWER
Yes
```

Note that none of the above failure modes are completely eliminated in the pipeline described in Section 2. Thus, human verification is required.

Despite struggling with the failure modes described above, there also exist cases where the models exhibit positively surprising and creative behaviors. We talk about some of them below.

**Thinking out of the box.**   Although rare, we observe instances where the models get creative while validating the question. Consider the question below

> **Example:** `Question:  A class of students is learning`
> `about combinatorics and geometry.  They are given`
> `a problem involving colored beads:  Red, Blue, and`
> `Green.  If they need to form a necklace with 8 beads`
> `such that no two adjacent beads have the same color`
> `and the necklace begins and ends with a bead of a`
> `different color, how many different necklaces can they`
> `create?  Each necklace is counted up to rotation and`
> `reflection (considering the necklace can be flipped`
> `over).`

When validating this question using prior iterations of the *question validation* prompt, which did not consist of the computational tractability check, the model output while validating the question consists of the following excerpt.

> **Example:**   *...This might introduce a significant challenge not solely due to the methodology's complexity but also due to the potential computational requirement, which may not be feasible in a standard test environment without tools. Furthermore, while the connection to practical geometry (reflective and rotational symmetry) and combinatorics (color patterning and adjacency constraints) is strong, the depth of understanding required to manually adjust for these symmetry considerations in a test question might be too intense or require more guided learning than a single evaluation question could provide....*

i.e, the model takes into consideration the fact that the question involves a lot of brute force computation, despite there being no explicit check for computation complexity in the prompt, and classifies

the question as invalid. We attribute such out of the box thinking behavior to the role-playing nature of our prompts. Our prompts consist of a math teacher evaluating the the fitness of the given question for being used for testing students' reasoning and analytical skills in a math exam. This leaves room open for the model to detect potential problems not explicitly accounted for in the prompts which might make the question unfit for being used for evaluation.

## A.2 CONSIDERATIONS FOR HUMAN-ANNOTATERS

Human annotators were tasked with double checking the validity of the question and the correctness of the solution. They were asked to look out for any of the failure modes discussed in Section 4. They were asked to check that the created question actually used the math skills it was supposed to exhibit and to improve the question with respect to readability, quality and difficulty. They were encouraged to suggest changes that make the problem harder to solve using automated tools while retaining easiness for the humans. We illustrate with an examples.

GPT-4 created the following question given the skill-tags `recursive_functions_and_sequences` and `multiplication_and_division`:

> **Example:** Original Question: Consider the sequence
> defined recursively by $a_1 = 1$ and $a_{n+1} = 2a_n + n$ for all
> $n \geq 1$. What is the product of the first five terms of
> this sequence?

An LLM can solve this by simple computation. The human modified the question so that solving the problem requires understanding the underlying pattern.

> **Example:** Modified Question: A sequence is defined
> recursively as follows: the first term $a_1$ is $2$, and
> for $n \geq 2$, $a_n = 2^{n-1} + n$. What is the logarithm (base 2)
> of the average of the first 50 terms of this sequence?
> Round down to the nearest integer.

For the modified question, one leading model mentioned calculation difficulties for the inability to give any answer, and another resorted to an incorrect numerical approximation that led to an incorrect answer.

Human annotators were also asked to go through the solutions carefully and correct or improve the solution for good questions if necessary. They were also asked to look out for questions that contain lot of enumeration, i.e. questions which are tedious and require significant amount of brute force computation. For such questions, the annotators were encouraged to reword them such that enumeration is not a feasible strategy below. For example, given below is an example of an enumerative question which was modified to avoid enumeration.

> **Example:** Original Question: Find the sum of the
> smallest prime divisor and the largest prime divisor
> of the number $N = 15^4 + 16^4$.
> Modified Question: Find the sum of the two smallest
> prime divisors of $23^{17} + 17^{17}$.

Models tend to adopt brute force approach on the original question calculating $15^4 + 16^4$. After rephrasing the models cannot use brute force on $23^{17} + 17^{17}$, instead being forced to check the divisors more analytically, in particular understanding of arithmetic modulo a prime.

## A.3 FURTHER EXPERIMENTAL DETAILS AND RESULTS

For open source LLMs, we use 2 80GB A100 GPUs and 72GB of RAM to run inference facilitated by vLLM (Kwon et al., 2023). We use 25 workers while querying GPT-4 Omni and GPT-4 Turbo and 2 workers for querying Claude-3 Opus and Claude-3.5-Sonnet.

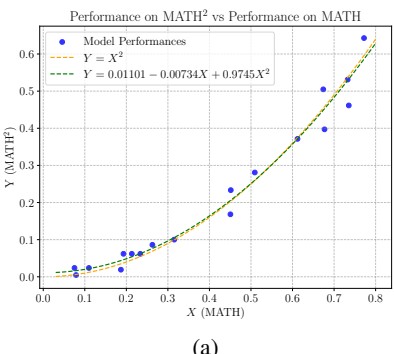

(a)

Figure 4: Relation between the performance of models on MATH$^2$ ($Y$) vs their performances on MATH ($X$). As can be seen from the plot, the performance on models on generated questions roughly follows a quadratic relation with the performance of those models on MATH. The best quadratic fit follows the relation: $Y = 0.01101 - 0.00734X + 0.9745X^2$. This may be explained by the fact that the questions in MATH$^2$ consist of two skills at a time, as compared to questions in MATH, which consist of one skills.

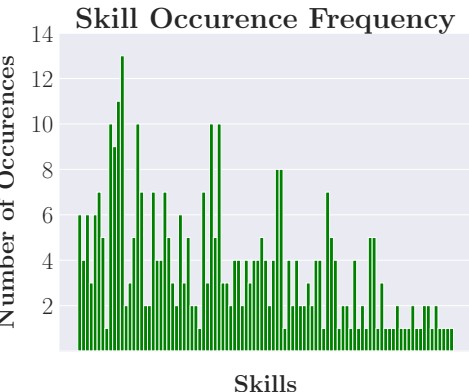

Figure 5: Shows the distribution of different skills extracted during the skill extraction process in the generated set of questions. The generated and human verified set of 210 questions consists of 93 skills out of the 114 skills extracted via the skill extraction process as described in Didolkar et al. (2024), Each question in the generated set represents two skills. Note that the distribution of skills is not uniform with there being multiple skills that are represented by one one question.

### A.3.1 EFFICIENCY AND COST OF THE QUESTION GENERATION PIPELINE

Below, we provide some statistics on the number of questions filtered out at different stages of the pipeline for different models. Note that these numbers are representative numbers, calculated on batches of data generated using each model. Questions in the MATH$^2$ dataset do not all necessarily belong to these batches.

**Success Rate of the AI Pipeline.** Table 4 we reports the number of questions filtered out during different stages of the AI pipeline (i.e., before the human verification step). Validation Step column reports the number of questions that were classified as "invalid" by the models in Step 4 (Question Validation) of the pipeline. Majority Agreement column reports the number of questions that were discarded because the final answers resulting from all 4 solution traces in Step 5 (Final Solution) of the pipeline were unique. Additionally, in our pipeline, we use regular expressions to extract the desired output from the rest of the response of the model at each stage. In some cases, the regex failed to extract the desired parts of the model response due to the model not following the specified output format. These numbers are reported in the Parsing Error column. The Total Rejected column sums

the aforementioned columns up and the Total Generated column contains the number of questions that were generated in Step 2 (Question Generation) of the AI pipeline.

Overall, GPT-4-Turbo turns out to be the most efficient model, in terms of the number of originally generated questions that made it to the end of the pipeline.

Table 4: Success rate of the AI pipeline for various models

| Model | Validation Step | Majority Agreement | Parsing Error | Total Rejected | Total Generated | Success Rate |
|---|---|---|---|---|---|---|
| GPT-4o | 850 | 345 | 48 | 1243 | 1972 | 36.97% |
| GPT-4 Turbo | 1958 | 748 | 64 | 2770 | 5115 | **45.84%** |
| Claude-3 Opus | 257 | 27 | 24 | 308 | 408 | 24.51% |
| Gemini-1.5-Pro | 935 | 229 | 16 | 1180 | 1434 | 17.71% |

**Success Rate of Human Verification.** Table 5 reports the number of questions annotated per model, and how many questions out of those made it to the dataset. The human annotators were asked to judge whether the questions (after any possible or necessary modifications), were good or a bit too easy. The questions marked "good" were included in the dataset.

Table 5: Human verification success rate comparison across models.

| Model | # of Questions Annotated | # of Questions Passed | Success Rate |
|---|---|---|---|
| GPT-4o | 28 | 3 | 10.71% |
| GPT-4 Turbo | 488 | 116 | 23.77% |
| Claude-3 Opus | 236 | 51 | 21.61% |
| Gemini-1.5-Pro | 61 | 40 | **65.57%** |

**Cost efficiency of the framework.** Table 6 reports an estimated cost of the data generation pipeline for each model. For each model, we calculate the average lengths of the input prompts and the generations (summing over all the steps in the pipeline) over 20 interactions. Next, we calculate the average cost for generating 1 question by using the formula:

$$\texttt{cost\_per\_question} = \texttt{avg\_input\_prompt\_length} \times \texttt{cost\_per\_input\_token} + \texttt{avg\_generation\_length} \times \texttt{cost\_per\_output\_token}$$

Next, we proceed to calculate the total cost for questions generated by the model using the formula

$$\texttt{total\_cost} = \frac{\texttt{cost\_per\_question} \times \texttt{num\_model\_questions\_in\_math}^2}{\texttt{human\_verification\_efficiency} \times \texttt{ai\_pipeline\_efficiency}}$$

where `human_verification_efficiency` and `ai_pipeline_efficiency` for the given model are taken as calculated in the previous two sections, and `num_model_question_in_math`[2] are stated in Section 3.1 of the paper. It is important to note that the result costs would be an estimate of the upper bound, since many of the rejected questions in the AI pipeline stage (Tables 4 and 5) are rejected in the question validation stage and thus the solutions for such questions are not generated in Final Solution generation stage, saving up on output generation costs.

Table 6: Cost and prompt length comparison for various models.

| Model | Avg. O/P Prompt Length | Avg. I/P Prompt Length | Cost Question | Total Cost |
|---|---|---|---|---|
| **GPT-4-Turbo** | 4614.85 | 133833.00 | $1.48 | $1575.60 |
| **GPT-4o** | 6080.95 | 135618.65 | $0.40 | $30.31 |
| **Claude-3 Opus** | 4066.70 | 134335.05 | $2.32 | $2233.88 |
| **Gemini-1.5-Pro** | 4851.85 | 136314.60 | $0.23 | $79.22 |
| | | | | **$3919.01** |

### A.3.2   SKILL PROPORTIONAL COMPARISON OF MATH$^2$ AND MATH

Figure 5 shows the distribution of different skills in MATH$^2$. To make a fairer comparison of MATH and MATH$^2$, and to show empirically that MATH$^2$ benefits from the composition of two skills at the same time as compared to MATH which consists of application of one skill at a time, we compare the performance of models on MATH$^2$ to the performance of models on a subset of MATH which has as similar skill distribution as MATH$^2$ (i.e. as shown in Figure 5). We form this subset by randomly sampling questions belonging to each skill in MATH. The subset consists of 4087 questions. Table 7 compares the performance of some models on MATH$^2$, MATH and the subset of MATH formed above. From the performance of the models, we can conclude that a subset of MATH with a similar distribution of skills is not just easier than MATH$^2$, but also MATH.

Table 7: Comparison of the performance of various models on MATH, MATH$^2$ and a subset of MATH which has a similar distribution of skills as MATH$^2$, as shown in Figure 5

| Model | MATH$^2$ (Y) | MATH skill proportional subset | MATH (X) |
|---|---|---|---|
| GPT-4 Omni | 64.29% | 79.28% | 77.21% |
| GPT-4 Turbo | 53.11% | 74.49% | 73.27% |
| Deepseek-math-7b-instruct | 16.83% | 45.60% | 45.05% |
| Mixtral-8x7B-Instruct | 10.00% | 31.98% | 31.52% |

### A.3.3   DIFFICULTY OF QUESTIONS GENERATED BY DIFFERENT MODELS

Out of the 210 questions, 116 questions were generated using GPT-4 Turbo, 3 using GPT-4 Omni, 51 using Claude-3 Opus and 40 using Gemini-1.5-Pro. We consider individual subsets of dataset wherein the questions were generated by GPT-4o, GPT-4-Turbo and Gemini-1.5-Pro and evaluate GPT-4O, GPT-4 Turbo, Claude-3 Opus and Gemini-1.5-Pro on these subsets. The results are shown in Table 8

Table 8: Performance of GPT-4 and Claude on questions generated using GPT-4 Turbo and Claude-3 Opus

| Subset | GPT-4 Omni | GPT-4 Turbo | Gemini-1.5-Pro | Claude-3 Opus |
|---|---|---|---|---|
| GPT-4 Turbo Subset | 57.76% | 54.31% | 43.47% | 40.52% |
| Claude-3 Opus Subset | 82.35% | 66.67% | 50.98% | 45.10% |
| Gemini-1.5-Pro Subset | 36.58% | 36.58% | 39.02% | 29.27% |

The results above show that the questions generated by Gemini-1.5-Pro ended up being significantly more difficult than the questions generated by other models.

### A.3.4   MODIFIED QUESTIONS VS NON-MODIFIED QUESTIONS

During the human verification process, the annotators were instructed to be on the look out for any errors in the questions and solutions generated by the models, and fix any lack of clarity, ambiguity, convoluted language, etc. in the generated questions which might confuse the model and reduce the "quality" of the questions. They were also instructed to look out for specific modifications which could make the questions more difficult. For further discussion on the human verification process, refer to Section A.2. In Table 9 we compare the performance of models on the questions which were modified against their performance on the questions which were not modified. We see that depending on the model being evaluated, they find either the modified or the unmodified questions more difficult.

### A.4   SKILLS USED FOR GENERATING THE QUESTIONS

Table 9: Performance of models on human modified and non-modified questions from MATH$^2$

| Model | MATH$^2$ Unmodified | MATH$^2$ Modified |
|---|---|---|
| GPT-4 Omni | 65.75% | 50.00% |
| GPT-4 Turbo | 58.90% | 59.37% |
| Claude-3.5-Sonnet | 47.94% | 50.00% |
| Gemini-1.5-Pro | 44.52% | 58.73% |

We use the same skills as those extracted in Didolkar et al. (2024). Table 10 lists the skills used during the question generation process.

Table 10: Skills used for question generation, as taken from Didolkar et al. (2024)

| Topic | Skills |
|---|---|
| Pre-Algebra | average_calculations, ratio_and_proportion, geometry, basic_arithmetic_operations, fractions_and_decimals, probability_and_combinatorics, multiplication_and_division, counting_and_number_theory, prime_number_theory, multiples_and_zero_properties, solving_linear_equation, circles, exponentiation_rules, perimeter_and_area |
| Algebra | combinatorial_operations_and_basic_arithmetic, function_skills, calculation_and_conversion_skills, solving_equations, inequality_skills, graph_and_geometry_skills, number_theory_skills, factoring_skills, complex_number_skills, sequence_and_series_skills, quadratic_equation_skills, geometric_sequence_skills, polynomial_skills, ratio_and_proportion_skills, logarithmic_and_exponential_skills, algebraic_manipulation_skills, distance_and_midpoint_skills, arithmetic_skills, exponent_and_root_skills, algebraic_expression_skills, function_composition_skills |
| Inter-Algebra | solving_inequalities, understanding_and_application_of_functions, inequality_solving_and_understanding, quadratic_equations_and_solutions, calculus_optimization_skills, polynomial_skills, understanding_and_applying_floor_and_ceiling_functions, summation_and_analysis_of_series, function_composition_and_transformation, sequence_and_series_analysis_skills, solving_system_of_equations, understanding_and_utilizing_infininte_series, recursive_functions_and_sequences, complex_number_manipulation_and_operations, understanding_ellipse_properties, complex_numbers_related_skills, simplification_and_basic_operations, graph_understanding_and_interpretation, understanding_logarithmic_properties_and_solving_equations, understanding_and_manipulation_of_rational_functions, properties_and_application_of_exponents, algebraic_manipulation_and_equations, prime_number_recognition_and_properties, absolute_value_skills |
| Geometry | understanding_circle_properties_and_algebraic_manipulation, other_geometric_skills, pythagorean_skills, quadrilateral_and_polygon_skills, triangle_geometry_skills, calculus_skills, 3d_geometry_and_volume_calculation_skills, circle_geometry_skills, area_calculation_skills, coordinate_geometry_and_transformation_skills, ratio_and_proportion_skills, trigonometry_skills, combinatorics_and_probability_skills, algebraic_skills |
| Number Theory | base_conversion, prime_number_theory, greatest_common_divisor_calculations, modular_arithmetic, solving_equations, number_theory, factorization, division_and_remainders, exponentiation, sequence_analysis, arithmetic_sequences, basic_arithmetic, polynomial_operations, understanding_of_fractions, number_manipulation |
| Precalculus | matrix_operations, geometric_series_comprehension, basic_trigonometry, vector_operations, coordinate_systems, trigonometric_calculations, complex_numbers, geometric_relations, calculus, algebra_and_equations, three_dimensional_geometry, arithmetic_operations, parametric_equations, sequences_series_and_summation, geometry_triangle_properties, geometry_and_space_calculation, determinant_calculation, geometry_transforms, complex_number_operations |
| Probability | probability_calculation_with_replacement, combinatorics_knowledge, probability_theory_and_distribution, combinatorial_mathematics, counting_principals, permutation_and_combinations, probability_concepts_and_calculations, calculating_and_understanding_combinations, number_theory_and_arithmetic_operations, factorials_and_prime_factorization, understanding_and_applying_combinatorics_concepts |

### A.5 EXAMPLE OUTPUTS FOR EACH STEP OF THE AI PIPELINE

In section, we present an example of a question-solution pair generation by giving the *extracted* output of each of the 5 steps in the AI pipeline. The skills used in this case are `sequence_analysis` and `polynomial_operations`

## Step 1: Skill Pair Validation

Let's break down the two skills and see how they relate:
**SKILL 1: sequence_analysis**

* **Focus:** Understanding patterns and relationships within sequences of numbers.
* **Key Concepts:** Arithmetic/geometric sequences, repeating patterns, modulo operations, identifying terms based on given rules.

**SKILL 2: polynomial_operations**

* **Focus:** Manipulating and solving equations involving polynomials.
* **Key Concepts:** Factoring, expanding, understanding roots, polynomial division, manipulating exponents.

**Comparison:**

While both skills fall under the umbrella of algebra, they address distinct mathematical concepts:

* **Nature of Objects:** Sequence analysis deals with ordered lists of numbers, while polynomial operations work with algebraic expressions involving variables and coefficients.
* **Problem-Solving Approaches:** Sequence analysis often involves pattern recognition and iterative thinking. Polynomial operations rely on algebraic manipulation and equation-solving techniques.
* **Applications:** Sequence analysis is common in number theory, discrete mathematics, and analyzing patterns in data. Polynomial operations are fundamental to algebra, calculus, and modeling relationships in various fields.

**Overlap:**

There might be rare instances where the two skills intersect. For example:

* A sequence could be defined by a polynomial formula, requiring polynomial operations to analyze its behavior.
* Finding the sum of a geometric series involves manipulating a polynomial expression.

However, these are specific scenarios and don't make the skills fundamentally the same.

## Step 2: Question Generation

The sequence $a_1, a_2, a_3, \ldots$ satisfies

$$a_n = \frac{n^3 + 2n^2 + n}{n^2 + 1}$$

for all positive integers $n$. Let $b_n$ be the product of the first $n$ terms of the sequence. Find the units digit of $b_{2023}$.

## Step 3: Attempted Solution

We can factor the numerator of $a_n$:

$$a_n = \frac{n(n^2 + 2n + 1)}{n^2 + 1} = \frac{n(n+1)^2}{n^2 + 1}.$$

Let $u_n$ be the units digit of $n$. Then the units digit of $n^2$ is the same as the units digit of $u_n^2$, and the units digit of $n + 1$ is the same as the units digit of $u_n + 1$. Hence, the units digit of $a_n$ is the same as the units digit of

$$\frac{u_n(u_n + 1)^2}{u_n^2 + 1}.$$

We build a table:

| $u_n$ | 0 | 1 | 2 | 3 | 4 | 5 | 6 | 7 | 8 | 9 |
|---|---|---|---|---|---|---|---|---|---|---|
| $\frac{u_n(u_n+1)^2}{u_n^2+1}$ | 0 | 1 | $\frac{12}{5}$ | $\frac{52}{10}$ | $\frac{100}{17}$ | $\frac{180}{26}$ | $\frac{294}{37}$ | $\frac{448}{50}$ | $\frac{648}{65}$ | $\frac{900}{82}$ |

We see that the units digit of $a_n$ cycles with period 10.

The units digit of $b_n$ depends only on the units digits of $a_1$, $a_2$, ..., $a_n$. Since 2023 divided by 10 has remainder 3, the units digit of $b_{2023}$ is the same as the units digit of $b_3 = a_1 a_2 a_3$. From our table, the units digit of $a_1 a_2 a_3$ is the same as the units digit of $1 \cdot \frac{12}{5} \cdot \frac{52}{10} = \frac{156}{25}$, which is $\boxed{6}$.

## Step 4: Question Validation

# REASONING

The question asks for the units digit of a product derived from a sequence. This necessitates analyzing the sequence's terms and their units digits' behavior. Let's break down the assessment:

1. **Single Answer Requirement:** The question satisfies this criterion. It clearly seeks a single output, the units digit of the product of the first 2023 terms of the sequence (represented by $b_{2023}$).

2. **Exact Answer Requirement:** The solution successfully arrives at a single, exact answer (6) for the units digit. It determines this by recognizing a repeating pattern of units digits within the sequence, eliminating the need for extensive calculations.

3. **Dual Skill Requirement:**
- **Sequence Analysis:** The core of the problem revolves around understanding the given sequence $a_n$, analyzing its terms, and identifying the cyclical behavior of the units digits. This aligns with the ""sequence_analysis"" skill, as demonstrated in the provided examples that involve identifying patterns, remainders, and modular arithmetic within sequences.
- **Polynomial Operations:** The problem requires manipulating the polynomial expression for $a_n$. Factoring the numerator and simplifying the expression are key to recognizing how the units digit of each term behaves. This demonstrates the application of ""polynomial_operations"" as needed. The complexity of polynomial operations is on par with the given examples for the skill, involving factoring and simplification to a more manageable form.

4. **Clarity and Completeness:** The question is well-defined. It provides the formula for the sequence $a_n$, clearly defines $b_n$ as the product of the sequence's terms, and specifies the desired output (units digit of $b_{2023}$).

5. **Computational Tractability:** The solution leverages pattern recognition based on the units digit cycle within the first ten terms. This avoids the need for calculating 2023 terms of the sequence, making it manageable for a timed exam without calculators.

6. **Relevancy of Information:** All information provided, the sequence formula, and the definition of $b_n$ are directly relevant to solving the problem. There are no extraneous details.

7. **Realism and Logic:** While not tied to a real-world scenario, the question presents a purely mathematical problem that is logically sound and internally consistent.

8. **Syntax and Grammar:** The problem is grammatically correct and uses clear mathematical notation, ensuring easy understanding.

**Overall, the question effectively combines the concepts of sequence analysis and polynomial operations to arrive at the solution. The difficulty level and knowledge required seem appropriate for assessing students' proficiency in both skills simultaneously.**

> # FINAL ANSWER
> Yes

## Step 5: Final Solution

We begin by factoring the numerator and denominator of $a_n$:

$$a_n = \frac{n(n^2 + 2n + 1)}{(n^2 + 1)} = \frac{n(n+1)^2}{n^2 + 1}.$$

We claim that $n^2 + 1$ and $(n + 1)^2$ are relatively prime for all positive integers $n$. To see this, suppose that $d$ is a common divisor of $n^2 + 1$ and $(n + 1)^2$. Then $d$ divides $(n + 1)^2 - (n^2 + 1) = 2n + 1$. Since $d$ divides $n^2 + 1$ and $2n + 1$, $d$ also divides $(2n + 1)^2 - 4(n^2 + 1) = 1$. Therefore, $d = 1$, so $n^2 + 1$ and $(n + 1)^2$ are relatively prime.

It follows that the fraction $\frac{n(n+1)^2}{n^2+1}$ is in lowest terms. Thus, the denominator of $b_n$ is

$$(1^2 + 1)(2^2 + 1)(3^2 + 1) \cdots (n^2 + 1).$$

The numerator of $b_n$ is

$$1(1 + 1)^2 \cdot 2(2 + 1)^2 \cdot 3(3 + 1)^2 \cdots n(n + 1)^2 = (n!)^2(n + 1)^2.$$

Therefore,

$$b_n = \frac{(n!)^2(n + 1)^2}{(1^2 + 1)(2^2 + 1)(3^2 + 1) \cdots (n^2 + 1)}.$$

We want to find the units digit of $b_{2023}$. Since 2023! contains factors of 2 and 5, the units digit of $b_{2023}$ is determined by the units digits of the factors in the denominator.

The units digits of $n^2 + 1$ cycle through the digits 2, 5, 0, 5, 2. Thus, the units digit of the product $(1^2 + 1)(2^2 + 1)(3^2 + 1) \cdots (2023^2 + 1)$ is the same as the units digit of $2^k 5^k$ for some positive integer $k$. This units digit is $\boxed{0}$.

A.6    SKILL COMPOSITION USING DIFFERENT MODELS

In this section, we provide examples of the responses of different models when asked to generate a question combining two different skills. We prompt GPT-4o, Claude 3 Opus, Gemini 1.5 Pro, and Llama-3-70B-Instruct and Llama-3-8B-Instruct to generate a question combining the skills `area_calculation_skills` and `algebraic_manipulation_skills`. We use the same prompt used in the **Question Generation** step (Step 2) of the proposed pipeline described in Section 2 (see Appendix A.7.2). Given below are the responses of each model to the prompt.

> ### GPT-4 Omni
>
> Sure! Here is a challenging math question that tests both area calculation skills and algebraic manipulation skills, formatted as requested.
>
> # QUESTION
> A rectangle $ABCD$ has vertices $A(0,0)$, $B(a,0)$, $C(a,b)$, and $D(0,b)$. Inside this rectangle, there is a right triangle $PQR$ where $P$ is at vertex $A$, $Q$ is on $BC$, and $R$ is on $CD$. The area of triangle $PQR$ is given by $\frac{1}{6}$ of the area of rectangle $ABCD$. Given that the coordinates of $Q$ and $R$ are $(a, kb)$ and $(ka, b)$ respectively, where $k$ is a constant, find the value of $k$.
>
> # SOLUTION
> First, let's find the area of the rectangle $ABCD$:
>
> $$\text{Area of rectangle } ABCD = a \cdot b$$

The area of the triangle $PQR$ is given as $\frac{1}{6}$ of the area of the rectangle $ABCD$:

$$\text{Area of triangle } PQR = \frac{1}{6} \cdot (a \cdot b) = \frac{ab}{6}$$

Next, we need to find the area of triangle $PQR$ using the coordinates of $P$, $Q$, and $R$.
The vertices of triangle $PQR$ are $P(0,0)$, $Q(a, kb)$, and $R(ka, b)$.

The formula for the area of a triangle with vertices at $(x_1, y_1)$, $(x_2, y_2)$, and $(x_3, y_3)$ is:

$$\text{Area} = \frac{1}{2} |x_1(y_2 - y_3) + x_2(y_3 - y_1) + x_3(y_1 - y_2)|$$

Substitute the coordinates of $P$, $Q$, and $R$ into the formula:

$$\text{Area of } PQR = \frac{1}{2} |0(kb - b) + a(b - 0) + ka(0 - kb)|$$

$$\text{Area of } PQR = \frac{1}{2} |0 + ab - ka^2 b|$$

$$\text{Area of } PQR = \frac{1}{2} |ab(1 - k)|$$

$$\text{Area of } PQR = \frac{ab}{2} |1 - k|$$

We know that the area of triangle $PQR$ is $\frac{ab}{6}$:

$$\frac{ab}{2} |1 - k| = \frac{ab}{6}$$

To solve for $k$, we can divide both sides by $\frac{ab}{2}$:

$$|1 - k| = \frac{1}{3}$$

This gives us two possible equations:

$$1 - k = \frac{1}{3} \quad \text{or} \quad 1 - k = -\frac{1}{3}$$

Solving these equations for $k$:

$$1 - k = \frac{1}{3} \quad \Rightarrow \quad k = 1 - \frac{1}{3} = \frac{2}{3}$$

$$1 - k = -\frac{1}{3} \quad \Rightarrow \quad k = 1 + \frac{1}{3} = \frac{4}{3}$$

Therefore, the possible values for $k$ are $\frac{2}{3}$ and $\frac{4}{3}$.

# DETAILS
The question involves area calculation by requiring students to find the area of a triangle within a rectangle, and algebraic manipulation by solving an equation involving a variable. The problem complexity is similar to the examples provided, ensuring that students are sufficiently challenged. The final answer consists of the value(s) of $k$, making it straightforward to evaluate the students' answers.

## Claude 3 Opus

Here is my attempt at generating 1 (non multiple choice) difficult question which tests both area_calculation_skills and algebraic_manipulation_skills:

# QUESTION
Consider an equilateral triangle ABC with side length 2a. Let points D, E, F be the midpoints of sides BC, CA, AB respectively.
Now construct squares ACDE, ABFG, BCHI externally on sides AC, AB, BC respectively.
Express the area of the shaded region (consisting of the equilateral triangle DEF and squares ACDE, ABFG, BCHI) in the form $ka^2$, where $k$ is a constant. Find the value of $k$.

# SOLUTION
Let the side length of the equilateral triangle ABC be $s = 2a$.
The area of equilateral triangle ABC is $\frac{\sqrt{3}}{4}s^2 = \frac{\sqrt{3}}{4}(2a)^2 = \sqrt{3}a^2$.

In an equilateral triangle, the distance from a vertex to the midpoint of the opposite side is $\frac{\sqrt{3}}{2}$ times the side length. So DE = EF = FD = $\frac{\sqrt{3}}{2}s = \sqrt{3}a$.

The area of equilateral triangle DEF is $\frac{\sqrt{3}}{4}(\sqrt{3}a)^2 = \frac{3\sqrt{3}}{4}a^2$.

The side length of each of the squares ACDE, ABFG, BCHI is equal to the side length of the equilateral triangle ABC, which is $s = 2a$.
So the area of each square is $s^2 = (2a)^2 = 4a^2$.

The total area of the shaded region is:
Area of DEF + 3 * Area of each square
$= \frac{3\sqrt{3}}{4}a^2 + 3(4a^2) = \frac{3\sqrt{3}}{4}a^2 + 12a^2 = (12 + \frac{3\sqrt{3}}{4})a^2$

Therefore, the area of the shaded region is $ka^2$, where $k = 12 + \frac{3\sqrt{3}}{4} \approx 15.1981$.

# DETAILS
This question combines area calculation of equilateral triangles and squares with algebraic manipulation of side lengths and areas. Key steps involve:

1) Calculating areas of equilateral triangles ABC and DEF in terms of side length $s$ and $a$ respectively. This tests area_calculation_skills.

2) Algebraically manipulating the expressions for side lengths DE, EF, FD and square side lengths in terms of $s$ and $a$. This requires algebraic_manipulation_skills.

3) Adding up the component areas algebraically and simplifying the expression in the form $ka^2$ to solve for constant $k$.

The question difficulty is calibrated to be similar to the provided examples. The solution is a single numerical constant $k$ which avoids ambiguity in student answers. The equilateral triangle and square constructions provide an interesting geometric visualization.

Let me know if you would like me to modify anything in the question or solution to better suit your needs! I'm happy to refine it further.

## Gemini 1.5 Pro

# QUESTION
A circle is inscribed in equilateral triangle $ABC$, and a smaller circle is externally tangent to this circle and to sides $AB$ and $AC$. If the side length of triangle $ABC$ is 4, what is the area of the smaller circle?

[asy] size(150); defaultpen(linewidth(0.8)); pair A=(0,2*sqrt(3)), B=(-2,0), C=(2,0), O=(0,sqrt(3)/3), P=(0,sqrt(3)); draw(A--B--C--A); draw(circle(O,sqrt(3))); draw(circle(P,sqrt(3)-1)); [/asy]

# SOLUTION
Let $O$ be the center of the larger circle, and let $P$ be the center of the smaller circle. Let $r$ be the radius of the smaller circle.

[asy] size(150); defaultpen(linewidth(0.8)); pair A=(0,2*sqrt(3)), B=(-2,0), C=(2,0), O=(0,sqrt(3)/3), P=(0,sqrt(3)); draw(A–B–C–A); draw(circle(O,sqrt(3))); draw(circle(P,sqrt(3)-1)); draw(O–P); draw(O–C); draw(P–C); label("$A$",A,N); label("$B$",B,SW); label("$C$",C,SE); label("$O$",O,S); label("$P$",P,NE); label("2",(1,0),S); label("$r$",(P+C)/2,E); label("1",(C+O)/2,SE); [/asy]

Since $O$ is the centroid of equilateral triangle $ABC$, $OC = \frac{2}{3} \cdot 2\sqrt{3} = \frac{4\sqrt{3}}{3}$. Also, $PC = r + 1$, so by the Pythagorean Theorem on right triangle $POC$,

$$r^2 + 2r + 1 = PC^2 = OC^2 - OP^2 = \frac{16}{3} - (r + \sqrt{3})^2.$$

This simplifies to $2r^2 + (2 + 2\sqrt{3})r - \frac{13}{3} = 0$. By the quadratic formula,

$$r = \frac{-(2 + 2\sqrt{3}) \pm \sqrt{(2 + 2\sqrt{3})^2 - 4 \cdot 2 \cdot -\frac{13}{3}}}{4} = \frac{-1 - \sqrt{3} \pm \sqrt{40}}{4}.$$

Since $r$ is positive,

$$r = \frac{-1 - \sqrt{3} + 2\sqrt{10}}{4}.$$

Then the area of the smaller circle is

$$\pi r^2 = \pi \left( \frac{-1 - \sqrt{3} + 2\sqrt{10}}{4} \right)^2 = \boxed{\frac{27\pi}{8} - \frac{13\sqrt{3}\pi}{8} + \frac{5\sqrt{30}\pi}{4}}.$$

# DETAILS
This question tests both area calculation skills (area of a circle) and algebraic manipulation skills (solving a quadratic equation with radicals). The problem is challenging because it involves nested radicals, but the steps are all fairly standard.

## Llama-3-70B-Instruct

Here is a difficult question that tests both area calculation skills and algebraic manipulation skills:

# QUESTION
A square with side length 4 is inscribed in a circle. A semicircle is drawn inside the square, with its diameter being one of the sides of the square. Find the area of the shaded region.

[asy]
size(100);
draw(unitcircle);
draw((-2,0)–(2,0)–(2,2)–(-2,2)–cycle);
filldraw((-2,0)–(2,0)–(2,2)–(-2,2)–cycle,lightgray);
[/asy]

# SOLUTION
The area of the shaded region is equal to the area of the square minus the area of the semicircle. The area of the square is $4^2 = 16$. The area of the semicircle can be found using the formula $\frac{1}{2}\pi r^2$, where $r$ is the radius of the semicircle. Since the diameter of the semicircle is 4, the radius is 2, so the area of the semicircle is $\frac{1}{2}\pi(2)^2 = 2\pi$. Therefore, the area of the shaded region is $16 - 2\pi$.

# DETAILS
This question requires students to calculate the area of a square and a semicircle, and then

subtract one from the other to find the area of the shaded region. It also requires algebraic manipulation skills to simplify the expression for the area of the semicircle. The answer is not a simple numerical value, but rather an expression involving $\pi$, which adds an extra layer of complexity to the question.

**Llama-3-8B-Instruct**

Here is a question that tests both area calculation skills and algebraic manipulation skills:

# QUESTION
Find the area of the region bounded by the curve $y = x^2 - 4x + 3$ and the $x$-axis, to the nearest hundredth.

# SOLUTION
The area of the region can be found by integrating the function $y = x^2 - 4x + 3$ with respect to $x$ from 0 to 2, which is $\int_0^2 (x^2 - 4x + 3)dx = [x^3/3 - 2x^2 + 3x]_0^2 = (8/3) - 8 + 6 - [0] = 2$. The area can also be found by using the formula for the area of a region bounded by a curve and the $x$-axis, which is $\int_0^2 y\,dx = \int_0^2 (x^2 - 4x + 3)dx = 2$. Therefore, the area is approximately 2 square units, to the nearest hundredth.

# DETAILS
This question tests both area calculation skills, as it requires the student to find the area of a region bounded by a curve and the $x$-axis, and algebraic manipulation skills, as it requires the student to integrate a function with respect to $x$.

Note: I've also reviewed the question and solution to ensure that they are accurate and clear for students.

A.7 PROMPT EXAMPLES

We detail the prompts used in the various stages of question generation and evaluation in the following sections. We present the skeleton of the prompts with placeholders for skill names, in-context exemplars, etc. in favor of keeping the prompt brief. The in-context exemplars and the skill exemplar repository are made available here.

A.7.1 SKILL PAIR VALIDATION

**Prompt for validating a pair of skills**

I have two basic mathematical skills described below, each followed by some sample question - solution pairs such that solving the questions involves the use of the particular skill.

SKILL 1: <skill_1>

Question 1: <skill_1_question_1>
Answer 1: <skill_1_solution_1>

Question 2: <skill_1_question_2>
Answer 2: <skill_1_solution_2>

Question 3: <skill_1_question_3>
Answer 3: <skill_1_answer_3>

SKILL 2: <skill_2>

Question 1: <skill_2_question_1>
Answer 1: <skill_2_solution_1>

Question 2: <skill_2_question_2>
Answer 2: <skill_2_solution_2>

Question 3: <skill_2_question_3>
Answer 3: <skill_2_solution_3>

I am going to use these two skills for framing a new question such that the question requires an expertise in both the skills in order to be solved, i.e. the question will compose these two skills. However, I do not want the two skills to be very similar, i.e., they should not mean the same thing. Go through the descriptions of the skills carefully. Based on your understanding of the skills, can you please tell me whether the two skills are essentially entirely the same or not? Think step by step and give a detailed explanation of your answer. The answer should begin with a prefix '# EXPLANATION '. Note that your understanding of the skills should not be restricted to the sample questions provided previously. They are just example questions. Use your own prior knowledge as well. End your response with a 'Yes' or 'No' answer to whether the skills are similar or not. This final answer should be on a new line and preceded by the prefix '# FINAL ANSWER '. Thank you very much!

## A.7.2 QUESTION GENERATION

### Prompt for question generation

I am a math teacher trying to create challenging math questions for smart students. I was wondering if you could give me 1 (non multiple choice) question which tests both the following skills: (<skill_1>, <skill_2>) Please also provide a brief solution. Then please look over the question and the solution, and fix any issues so that my students do not get frustrated. This being a math exam, the answers should either be exact, or if not possible, then the question should clearly say the answer is only expected to be approximately correct. Further, for ease of evaluating the students' answers, the question should ask for a single final result.This process is difficult so I am attaching two sample conversations where (Agent) is an AI agent and (Query) is teacher feedback. The conversations revolve around framing such mathematical reasoning questions and using them for evaluating students. These should give you some idea of the expectations and the potential difficulties involved in this task. I am also giving three example question - answer pairs for both <skill_1> and <skill_2> skills, such that the example questions test the corresponding skill. Please ensure that the complexity / difficulty of application of <skill_1> and <skill_2> skills in the generated question is similar to the complexity / difficulty of the skills in the example questions. Please format your output as

'# QUESTION
<question>

# SOLUTION
<solution>

# DETAILS
<all other text>'

SKILL 1: <skill_1>

Question 1: <skill_1_question_1>
Answer 1: <skill_1_solution_1>

Question 2: <skill_1_question_2>
Answer 2: <skill_1_solution_2>

Question 3: <skill_1_question_3>
Answer 3: <skill_1_solution_1>

SKILL 2: <skill_2>

Question 1: <skill_2_question_1>
Answer 1: <skill_2_solution_1>

Question 2: <skill_2_question_2>
Answer 2: <skill_2_solution_1>

Question 3: <skill_2_question_3>
Answer 3: <skill_2_solution_3>

# CONVERSATION 1
<agent_convo_1>

# CONVERSATION 2
<agent_convo_2>

### A.7.3 ATTEMPTED SOLUTION

Prompt for solution attempt. Note that we instruct the model to take a defeatist approach towards solving the question

**Prompt for solution attempt**

You are a professional math teacher and you are given a question which is supposed to test the analytical and mathematical reasoning abilities of your students. You are supposed to provide a solution to the given question. However, the question may be flawed. For example, it might have problems like question being unsolvable using the information provided, question being self-contradictory, the final answer being computationally intractable, the question being ambiguous and confusing, question having multiple possible interpretations, etc., which you may encounter while solving the problem. This question being used for evaluating students in math, the question should ideally have a single, exact answer, with no room for any deviations due to factors such as approximations, rounding errors, etc., unless explicitly specified in the question. Problems such as the ones described above, would prevent the students from solving the question properly, and thus, any question with either of these problems is unfit for testing the students. If you encounter any such problems, stop the solution right there and explain the problems. For example, if you encounter the need to make any approximations or rounding which is not specified in the question, stop solving the question along with the reason. You do not need to solve the question further once you encounter any such problem. If you do not encounter any such problem, solve the question to achieve the single exact answer which the question asks for.

# QUESTION
<question>

### A.7.4 QUESTION VALIDATION

Note that how in the first paragraph, the names of the two skills are mentioned even time instead of using referential phrases. This is done to address the *lost in the middle* problem

**Prompt for validating the questions**

You are a professional math teacher. You want to evaluate the analytical and mathematical reasoning abilities of your students in a math exam. The students are supposed to sit in an examination hall and solve the questions within a given time limit, without access to any computational devices. The evaluation is designed to test the students' expertise in using two given mathematical skills simultaneously, namely <skill_1> and <skill_2>. This is achieved by asking them to solve a question that necessitates expertise in both <skill_1> and <skill_2> skills, to be solved completely. Since evaluating the students is a critical task allowing very little margin for any error in the process, it is very important to ensure that the questions used for evaluating are high quality and fit for being used to evaluate the students. You need to carefully review the question and a given attempt at solving it, and ensure that the question is of high quality and fit to assess students. In order to do this, you should check the quality of the question with respect to several criteria, such as:

- Single Answer Requirement: The question should ask for one and only one final result. It should not request multiple distinct answers or pieces of information.
- Exact Answer Requirement: It should be possible to achieve one,exact answer to the question, without the need of making any approximations or assumptions whatsoever, unless explicitly specified in the question. There should be no margin for the students to arrive at any other possible answer due to things like rounding errors, etc.
- Dual Skill Requirement: The question must require rigorous expertise in both a: a) '<skill_1>' and b) '<skill_2>', for resolution. Application of both <skill_1> and <skill_2> and their subskills should be, necessary and contribute directly to obtaining the final answer; <skill_1> and <skill_2> skill should be applicable separately and critically during the problem-solving process. You are also given three example question - answer pairs for both <skill_1> and <skill_2> skills in order to help you better understand the meaning of each skill. Please carefully review the question and its attempted solution, paying close attention to how well it aligns with the examples provided for each skill. Consider the depth and breadth of knowledge demonstrated in the examples. The complexity / difficulty of application of both <skill_1> and <skill_2> in the question should be similar or greater than the complexity / difficulty of <skill_1> and <skill_2> in the example question-answers given for that respective skill.
- Clarity and Completeness: The question should be unambiguous and contain all the information necessary to complete the solution. Any required assumptions not common knowledge should be explicitly stated. Check for any ambiguity that might confuse students. Carefully go through the solution to check if it makes any assumption or approximation in order to solve the question.
- Computational Tractability: Since the students are supposed to solve the questions within a given time limit and without access to any computational devices such calculators, computer, mobile phones, etc., you must ensure that the question is computationally tractable and all the computations involved can be done by hand in a limited amount of time.
- Relevancy of Information: The question should not have any extra details that do not contribute to the solving of the problem.
- Realism and Logic: The question should involve realistic scenarios or hypotheses with logically consistent data. The specified operations and the contextual setup should reflect plausible mathematical situations. (e.g., positive amounts for transactions, integers for counts).
- Syntax and Grammar: The question must be grammatically correct and clearly written to prevent misinterpretation.
- etc. (any other problems which you think make the question not fit for being used for evaluating the students)
Your task is to give a 'Yes' or 'No' assessment, indicating whether the question is high quality and suitable for evaluating the students on simultaneous application of the skills <skill_1> and <skill_2>. Provide thorough reasoning for your assessment based on the conditions mentioned above and any other relevant analytical points concerning mathematical reasoning and problem-solving. Your response should be structured as follows:

# REASONING
<Your detailed analysis justifying your decision>

# FINAL ANSWER
<'Yes' or 'No'. No other text should be present in this section>

Ensure to review the combination of skills intended for assessment, and check the logical flow and mathematical correctness from the question's setup to the solution's conclusion. Look out for any problems in the question which are pointed out in the attempted solution. Account for all the potential pitfalls such as logical inconsistencies, unnecessary complexity, or insufficient detail that may obstruct the clarity or solvability of the question. Given below are the two skills and some example question-answer pairs for the two skills. This process is difficult so I am attaching a few sample conversations where (agent) is an AI agent who is trying to verify the questions and (query) is teacher feedback. This should give you some idea of potential difficulties in this task. This is followed by the question which you need to check (preceded by '# QUESTION TO BE CHECKED') and its attempted solution (preceded by '# SOLUTION ATTEMPT').

SKILL 1: <skill_1>

Question 1: <skill_1_question_1>
Answer 1: <skill_1_solution_1>

Question 2: <skill_1_question_2>
Answer 2: <skill_1_solution_2>

Question 3: <skill_1_question_3>
Answer 3: <skill_1_solution_3>

SKILL 2: <skill_2>

Question 1: <skill_2_question_1>
Answer 1: <skill_2_solution_1>

Question 2: <skill_2_question_2>
Answer 2: <skill_2_solution_2>

Question 3: <skill_2_question_3>
Answer 3: <skill_2_solution_3>

# CONVERSATION 1
<validation_exemplar_1>

# CONVERSATION 2
<validation_exemplar_2>

......

# CONVERSATION 6
<validation_exemplar_6>

# QUESTION TO BE CHECKED
<question>

# SOLUTION ATTEMPT
<solution>

Thank you very much!

### A.7.5  FINAL SOLUTION

For the final solution, we make use in-context exemplars from MATH (Hendrycks et al., 2021) as opposed to the attempted solution step.

**Prompt for the final solution**

I have two basic mathematical skills described below, each followed by some sample question - solution pairs such that solving the questions involves the use of the particular skill in order to be solved.

SKILL 1: <skill_1>

Question 1: <skill_1_question_1>
Answer 1: <skill_1_solution_1>

Question 2: <skill_1_question_2>
Answer 2: <skill_1_solution_2>

Question 3: <skill_1_question_3>
Answer 3: <skill_1_solution_3>

SKILL 2: <skill_2>

Question 1: <skill_2_question_1>
Answer 1: <skill_2_solution_1>

Question 2: <skill_2_question_2>
Answer 2: <skill_2_solution_2>

Question 3: <skill_2_question_3>
Answer 3: <skill_2_solution_3>

Go through the descriptions of the skills carefully. Now, here is a new question such that the question requires an expertise all both the skills in order to be solved. That is, the question composes these two skills

QUESTION: <question>

Based on your understanding of the skills, can you please solve the question accurately? Think step by step and explain the solution. Finally, end your response by stating the final numerical answer obtained using the solution. Note that your understanding of the skills should not be restricted to the sample questions provided in their description. They are just example questions. Use your own prior knowledge as well. The explanation of your solution and the final numerical answer should each be on a new line, and should be preceded by the prefixes '# SOLUTION ' and '# ANSWER ' respectively. Thus, your response should be in the format:

'# SOLUTION
<solution>

# ANSWER
<final_answer; no other text should be present in this section>'.

Thank you very much!

## A.7.6 EVALUATION

**Prompt given to the GPT-4 for evaluating the model's solution**

You are a professional math teacher and are tasked with evaluating your students on a math exam. You are will be given a question, the correct solution to the question and the student's solution. You need to tell me whether the student solved the question correctly, thus matching the answer obtained by the correct solution. Think step-by-step and give a detailed explanation of your answer. At the end, give a 'Yes' or 'No' answer to whether the student's solution is correct. Your output should be in the following format:

# STEP BY STEP EXPLANATION
<detailed explanation of your thought process>

# CORRECTNESS
<'Yes' if the student's solution is correct. 'No' otherwise. This section should not contain any other text>

Here are the question, correct solution to the question and the student's solution:

QUESTION: <question>

CORRECT SOLUTION: <correct_solution>

STUDENT'S SOLUTION: <student's_solution>

