# OpenReview forum: "AI-Assisted Generation of Difficult Math Questions"
_ICLR.cc/2025/Conference — Submitted to ICLR 2025_

### Official Review · Reviewer_KurQ · 2024-11-01

**Soundness:** 3
**Presentation:** 3
**Contribution:** 4
**Rating:** 6
**Confidence:** 3

**Summary:**

This paper introduces a framework for generating challenging math questions using Large Language Models (LLMs). The process involves extracting skills from existing math questions and combining two different skills to construct new questions. Their curated dataset, "MATH^2," consistently shows lower performance compared to the original MATH dataset, exhibiting a square relationship with MATH accuracy.

**Strengths:**

The paper is well-structured and clearly written.

The motivation to use LLMs for generating more difficult questions to evaluate these models is a promising direction. This paper represents a good attempt.

The performance of MATH^2 demonstrates that it is a challenging dataset, which can benefit research on math generalization and evaluation.

**Weaknesses:**

The dataset construction still requires human intervention to modify or verify correctness, which is not scalable.

A main concern is the "out-of-distribution" aspect. If we simply combine two questions from the MATH dataset, solving both questions correctly implies a squared accuracy,(it is not an out-of-distribution scenario). Therefore, as the MATH^2 achieve similar performance, does MATH^2 truly represent an out-of-distribution dataset?

**Questions:**

See the Weakness

---

> ### Author Response · Authors · 2024-11-20
> **Response to the reviewer's feedback**
>
> We thank the reviewer for taking the time to go through our paper and providing useful comments. We attempt to answer the comments raised by the reviewer below
>
> ### OOD Claim
> We argue that despite the fact that MATH^2 achieves a performance similar to combining two questions in the MATH dataset, it is still OOD. One major reason is that MATH has seven domains---pre-algebra, algebra, inter-algebra, gemotery, number theory, precalculus and probability, each consisting of questions belonging to the same topic. However, while generating MATH^2 questions, we randomly sample pairs of skill from across all these 7 domains, not just within each domain. Since these questions combine domains, they can be considered OOD.
>
> It is important to note here that this property of being skills being sampled from across different domains is not sufficient to ensure that the generated questions are more difficult than the typical MATH question. The reason is that even if the skills are fairly different from each other, it could be that the generator is simply incompetent at combining them to create an interesting and difficult question.
>
> We are happy to provide any further clarifications and answer any further questions that the reviewer might have.

---

> > ### Comment · Reviewer_KurQ · 2024-11-26
> >
> > Thanks for authors's response. I will keep my score unchanged.

---

> > > ### Author Response · Authors · 2024-12-01
> > >
> > > Dear reviewer,
> > > We hope we addressed all of your concerns in our rebuttal. Please let us know if you have any further concerns. We would happy to engage in further discussion in order to address those.
> > >
> > > If all of your concerns have been addressed, we kindly ask you to reconsider your assessment as the discussion period nears the end.

---

### Official Review · Reviewer_SXeA · 2024-11-02

**Soundness:** 3
**Presentation:** 2
**Contribution:** 2
**Rating:** 3
**Confidence:** 4

**Summary:**

The paper introduces a framework for generating challenging mathematical questions using a hybrid approach that combines LLMs with human verification. Recognizing the limitations of both human-generated and purely LLM-generated math problems, the proposed method aims to address the need for diverse and difficult questions. It uses LLMs to generate questions requiring the combination of two distinct mathematical skills and employs a multi-turn interaction system to refine these questions. Human annotators then review and adjust the questions to enhance clarity and difficulty. This process generated the "MATH^2" dataset, which proved to be more challenging for existing models than the baseline MATH dataset and also served as effective in-context training exemplars, thereby enhancing LLM performance on other datasets.

**Strengths:**

- The paper is well-structured and easy to follow
- The example generation pipeline can be extended to other structured reasoning domains and could be valuable to the research community
- Experimental results show that the generated examples are more challenging for existing models than the baseline MATH dataset and also served as effective in-context training exemplars

**Weaknesses:**

- The description of step 5 in Section 2 lacks clarity. The authors should provide more detail about the human re-validation process. For instance, do the human validators create their own solutions first and then compare them with those generated by the model? Additionally, it would be helpful to know the qualifications of these human experts, as the MATH dataset poses significant challenges, even for college-level students

I list some minor concerns in the questions section

**Questions:**

- Why aren’t human experts involved in the steps of solution attempts and question validation? Could this result in challenging but valid examples being filtered out?
- The authors might consider adding a comprehensive, step-by-step example for each stage in Section 2. This could significantly enhance readers' understanding of the pipeline’s specific implementation details and workflow, making it easier to grasp the mechanisms behind each stage of the generation process.
- How do different base LLMs in the pipeline impact generation performance? For instance, are models with stronger reasoning capabilities more likely to produce "valuable" examples?
- Will the model perform better on questions it generates itself compared to those produced by other base LLMs in the pipeline?
- What is the benchmark performance of human experts on the MATH2 dataset?
- What is the estimated time to generate a single example of similar difficulty to those in the MATH dataset under a human-in-the-loop setting?

---

> ### Author Response · Authors · 2024-11-20
> **Response to the reviewer's feedback**
>
> We would like to thank the reviewer for taking the time to go through our paper and providing helpful feedback. We attempt to address some of the concerns and questions raised by the reviewer below.
>
> ### More details about the human verification step
> The human annotators review the solutions generated by the model and modify them in order to correct them or improve their clarity. They do not come up with their own solutions and compare them with solutions generated by the model. The statistics of these modifications can be found in Section 3.1, Line 314. More details about the instructions provided to the human annotators are discussed in Appendix A.2.
>
> ### Qualifications of Human Annotators
> All the annotators involved in the human verification process are graduate students with backgrounds in Computer Science and Mathematics, ranging from undergraduate to graduate levels.
>
> ### More involvement of human annotators in the pipeline
> We aim to reduce the involvement of human annotators in the process of question generations. Involving human annotators in other steps of the pipeline such as question validation, although helpful in ensuring that good questions do not get filtered out, would lead to more human effort and thus, less scalability. At the end of the AI part of the question generation pipeline, humans verify the question **as well as the solution**. Considering this, humans can be seen as being involved in the solution-generation step.
>
> ### Addition of examples for each step of the pipeline
> We thank the reviewer for this useful suggestion. While we discuss examples of generated questions and validation step in Appendix A.2, A.3, and A.4, we will add a dedicated section on providing examples for each step to the paper.
>
> ### Impact of different LLMs on the generation performance
> Qualitatively, we observe that different LLMs seem to be good at different aspects of the pipeline. There is no correlation between the reasoning performance and the quality of the generation questions. While GPT-4o is the best reasoner (judging by the performance of models on different datasets), it is not a very good generator, which can be seen in the fact that only 3 questions out of the 210 questions were generated by GPT-4o. GPT-4-Turbo on the other hand is better than GPT-4o at generating good questions. Gemini-1.5-Pro however is notably good (the best among the models tested) at generating difficult questions. GPT-4o is good at generating solutions. Claude-3 Opus seemed to be doing well at the question validation step. Some empirical evidence of the above claims can be observed in Table 5, where the models perform worst on the subset of questions generated by Gemini-1.5-Pro indicating Gemini generated the most difficult questions (ignoring the counterfactual of human modification). On the other GPT-4o performs the best in solving questions across different subsets.
>
> ### Performance of models on self-generated questions
> We do not observe any positive correlation between the models and their performance on questions generated by themselves. We demonstrate this empirically in Table 5 (Appendix A.3.2). GPT-4o performs the best on different subsets of the data, where the subsets are determined according to the model that generated the question.
>
> We are happy to answer any further questions that the reviewer might have

---

> > ### Author Response · Authors · 2024-11-21
> > **Response Continued**
> >
> > ### Performance of humans MATH^2
> > The skills involved in the formulation of MATH^2 are AP level (i.e. high school level) skills and thus, MATH^2 is still AP-level math benchmark. So, human performance would be 100%. However, the questions take humans longer to solve (i.e. require longer reasoning).
> >
> > We would like to metion that the goal of the paper is to create harder questions that still involve the same math material. Our work shows that even on this material current models have gaps in understanding, once you we 2 skills per question. As models improve in the future, the proposed approach could be used to create more difficult evaluations based on more difficult math topics (i.e. more difficult than AP-level math).
> >
> > ### Time taken for generating MATH level questions
> > In subsequent work, we have focused on generating MATH-level questions. While we have not measured the exact time taken for generating these questions, we do observe that due to the questions being relatively simple (as compared to MATH^2), the question validation (Step 4), the final solution generation (Step 5) as well as the human verification steps have a significantly higher success rate, thus leading to the entire process taking significantly lesser time.
> >
> > We would be happy to engage in any further discussion with the reviewer and provide any further clarifications required.

---

> > > ### Comment · Reviewer_SXeA · 2024-11-28
> > >
> > > Thank you for your response! I'm inclined to give a score between 3 and 5 after considering the author's reply.

---

> ### Author Response · Authors · 2024-11-29
>
> Dear Reviewer,
> We appreciate your willingness to increase the scores. We believe there is no rating between 3 and 5 for ICLR 2025.
>
> However, please let us know if you have any further questions or concerns regarding the paper. We would be very happy to engage in further discussion to address them.

---

> > ### Author Response · Authors · 2024-12-01
> > **Gentle Reminder**
> >
> > Dear reviewer,
> > We hope that most of your concerns have been addressed. We would be happy to engage in further discussions if any of your concerns remain unaddressed.
> >
> > If most of your concerns have been addressed, we would appreciate it if you could reconsider your assessment as the discussion period nears its end

---

### Official Review · Reviewer_nXDa · 2024-11-03

**Soundness:** 2
**Presentation:** 3
**Contribution:** 2
**Rating:** 5
**Confidence:** 5

**Summary:**

This paper proposes a human-AI collaborative framework for generating challenging mathematical problems. The approach begins by identifying core skills from existing math questions and then prompts LLMs to create difficult questions that combine pairs of these skills.
The generated questions are then solved by the models, with the framework filtering out invalid ones based on QAs before proceeding to further human annotation. This process results in the Math$^2$ dataset, which includes 210 difficult questions. Evaluation on Math$^2$ reveals that model performance significantly declines, approximately in proportion to the square of the performance on the original MATH dataset. The results also suggest that these new questions can serve as more effective in-context example.

**Strengths:**

1. This work focuses on an AI-assisted framework for generating harder questions that models themselves struggle to solve. This is potentially of interest to the community, as many existing evaluation benchmarks are becoming saturated, often only reflecting model performance in overfitted domains.
2. The proposed framework results in the creation of a more difficult dataset, MATH$^2$, offering a new benchmark for evaluating mathematical reasoning abilities.
3. The content is clear and easy to follow. The experiments provide comprehensive evaluations across multiple models, demonstrating that the questions are indeed challenging. Additionally, the correlation between performances on MATH and MATH$^2$ is an interesting finding.

**Weaknesses:**

1. One major weakness is the generalizability of the proposed framework. Around 66% of AI filtered QAs need further human editing. This reliance on human annotation may impact the scalability and generalizability of the framework.

2. As an evaluation benchmark, the dataset is relatively small and lacks comprehensive coverage. Line 412 mentions that the dataset has not covered all possible skills. Given that the questions are initially generated automatically, it would be beneficial for the dataset to cover a broader range of skill sets.

3. The data generation framework is not resource efficient due to the high number of API calls involved. It would be helpful if the authors could provide an analysis of the resource costs for creating the dataset.

**Questions:**

1. Line 253: It is stated that if all the answers obtained are unique, the question is discarded. Could this indicate that some questions are too difficult to solve, leading to the discarding of extremely challenging questions?
2. The remaining questions are extremely few, so I am curious how many questions are filtered out at each step shown in Figure 1. What percentage of questions pass the filtering process?
3. Out of the 210 questions in MATH$^2$,what is the accuracy split between questions generated directly by the framework (210-139=71) and those further modified by human annotators (139)? Is the performance lower on the human-modified subset?
4. Line 75-76: Missing references for the findings
5. Line 322-323: Missing references for the models mentioned

---

> ### Author Response · Authors · 2024-11-20
> **Response to the reviewer's feedback**
>
> We thank the reviewer for going through our paper and providing us with useful feedback. Below, we attempt to address some of the questions and concerns raised by the reviewer.
>
> ### Scalability and Generalizability of the proposed framework
> We would like to argue that even with a 66% modification rate, the proposed framework would still be much more efficient than the process of manual question-solution pair creation by humans. The latter would involve steps such as (a) finding human experts for the particular domain (b) the human experts coming up with challenging questions that combine (c) the human experts coming up with correct solutions to the challenging questions; making the entire process very laborious and in-efficient. Compared to this, having human annotators review challenging questions and solutions already generated by LLMs is much more efficient. We would further like to argue that the modification rate would depend on the quality of the model generating the questions. During the question generation process, we observed that certain models are better at coming up with difficult questions than others, certain models are better at discriminating between valid and invalid questions than others and certain models are better at solving questions than others. More specifically, we observed that Gemini-1.5-Pro is good at coming up with challenging questions, Claude-3 Opus is good at validating questions and GPT-4o is good at solving questions. Thus, a pipeline wherein different models are used in different stages of the pipeline according to their specialty, would have a much lower modification rate. Another interesting avenue that can be explored is training specialist question-solution generation models, similar to as done in [1], but for generating evaluation data. Such specialist models, in addition to having a much lower post-generation modification rate, can also be obtained by finetuning smaller open-source models, thus removing the dependence of the proposed pipeline on proprietary models. We leave the ideas discussed above for future work.
>
> Finally, our human raters uniformly agreed that having the model answers (whether right or wrong) during the annotation process was quite helpful in efficiently verifying correctness, and if needed, supplying corrections to the answers. This is similar to other known settings of AI-assisted verification such as debates.
>
> ### Size and skill coverage of the dataset
> We agree with the reviewer on this point. The dataset discussed in the paper is meant to show the effectiveness of the proposed pipeline. We are working on expanding the dataset further.
>
> ### Number of questions filtered out per stage
>
> In the table given below, we report the number of questions filtered out during different stages of the AI pipeline (i.e. before the human verification step). **Validation Step** column reports the number of questions that were classified as "invalid" by the models in Step 4 (Question Validation) of the pipeline. **Majority Agreement** column reports the number of questions that were discarded because the final answers resulting from all 4 solution traces in Step 5 (Final Solution) of the pipeline were unique. Additionally, in our pipeline, we use regular expressions to extract the desired output from the rest of the response of the model at each stage. In some cases, the regex failed to extract the desired parts of the model response due to the model not following the specified output format. These numbers are reported in the **Parsing Error** column. The **Total Rejected** column sums the aforementioned columns up and the **Total Generated** column contains the number of questions that were generated in Step 2 (Question Generation) of the AI pipeline.
>
> Overall, GPT-4-Turbo turns out to be the most efficient model, in terms of the number of originally generated questions that made it to the end of the pipeline.
>
> | Model             | Validation Step | Majority Agreement | Parsing Error | Total Rejected | Total Generated | Success Rate |
> |-------------------|-----------------|--------------------|---------------|----------------|-----------------|--------------------|
> | GPT-4o            |        850      |          345       |     48        |       1243     |     1972        |       36.97%       |
> | GPT-4 Turbo       |       1958      |       748          |     64        |        2770    |  5115           |   **45.84%**       |
> | Claude-3 Opus     |       257       |        27          |     24        |    308         |     408         |   24.51%           |
> | Gemini-1.5-Pro    |         935     |     229            |      16       |       1180     |     1434        |    17.71%           |
>
> We plan to add these numbers to the next update of the paper (which will be before the end of the discussion period)

---

> ### Author Response · Authors · 2024-11-20
> **Response continued**
>
> ### Efficiency of Human Verification
> We would like to clarify that out of the many questions generated using different models and the proposed pipeline, we construct a dataset of only 210 questions due to limited resources in terms of human annotators. Nevertheless, we report below the figures for the number of questions filtered out in each step of the pipeline for different models. Note that these numbers are an estimate since they were calculated over different, independent representative batches of the data (including that for the human verification process) which may not correspond exactly to the data batches that questions in the MATH^2 dataset belong to.
>
> The table given below reports the number of questions annotated per model, and how many questions out of those made it to the dataset. The human annotators were asked to judge whether the questions (after any possible or necessary modifications), were good or a bit too easy. The questions marked "good" were included in the dataset.
>
> | Model          | # of questions annotated | # of questions passed | Success Rate |
> |----------------|--------------------------|-----------------------|--------------|
> | GPT-4o         |     28                   |          3            |      10.71%  |
> | GPT-4 Turbo    |        488               |          116          |      23.77%  |
> | Claude-3 Opus  |      236                 |            51         |       21.61% |
> | Gemini-1.5-Pro |        61                |              40       |  **65.57%**  |
>
> We plan to add these numbers to the next update of the paper (which will be before the end of the discussion period)
>
>
> ### Cost efficiency of the framework
> Below, we report an estimated cost of the data generation pipeline for each model. For each model, we calculate the average lengths of the input prompts and the generations (summing over all the steps in the pipeline) over 20 interactions. Next, we calculate the average cost for generating 1 question by using the formula:
>
> `cost_per_question = avg_input_prompt_length * cost_per_input_token + avg_generation_length * cost_per_output_token`
>
> Next, we proceed to calculate the total cost for questions generated by the model using the formula
>
> `total_cost = (cost_per_question * num_model_questions_in_math^2) / (human_verification_efficiency * ai_pipeline_efficiency)`
>
> where `human_verification_efficiency` and `ai_pipeline_efficiency` for the given model are taken as calculated in the previous two sections of the response, and `num_model_question_in_math^2` are stated in Section 3.1 of the paper. It is important to note that the result costs would be an estimate of the upper bound, since many of the rejected questions in the AI pipeline stage (see Section **Success Rate of the AI Pipeline**) are rejected in the question validation stage and thus the solutions for such questions are not generated in Final Solution generation stage, saving up on output generation costs.
>
> As discussed previously, this cost can be further optimized by using different models at different stages of the pipeline and/or training specialist math problem generation models.
>
> | Model               | Avg. Output Prompt Length | Avg. Input Prompt Length | Cost per Question| Total Cost |
> |---------------------|---------------------------|--------------------------|------------------|------------|
> | **GPT-4-Turbo**     | 4614.85                   | 133833                   | $1.48            | $1575.60   |
> | **GPT-4o**          | 6080.95                   | 135618.65                | $0.40            | $30.31     |
> | **Claude-3 Opus**   | 4066.7                   | 134335.05                 | $2.32            | $2233.88   |
> | **Gemini-1.5-Pro**  | 4851.85                  | 136314.6                  | $0.23            | $79.22     |
>
>
> Overall Cost = **$3919.01**
>
> We plan to add these numbers to the next update of the paper (which will be before the end of the discussion period)
>
> ### Model obtaining unique answers for difficult questions
> There can be several reasons for a model obtaining unique answers in each trial in the solution generation step, such as the question being ambiguous, containing insufficient information, being computationally intractable, etc. Another possible reason as correctly pointed out by the reviewer is that the question is too difficult for the model to solve. In such cases, a good question would get screened out. We identify this as one of the limitations of the pipeline. This limitation should become less prominent with the development of better reasoners.
>
> ### Accuracy split between modified and non-modified questions
> We discuss this experiment in Appendix A.3.3 in Table 6.
>
> ### Missing references
> We thank the reviewer for pointing these out. We will add the references in the next update to the paper.
>
> ### References
> [1] Ding et al., 2024. Unleashing Reasoning Capability of LLMs via Scalable Question Synthesis From Scratch

---

> > ### Comment · Reviewer_nXDa · 2024-11-25
> >
> > Thank you for your detailed response and for providing additional analysis.
> > After considering the information provided regarding the costs and the filtering success rate, I remain somewhat unconvinced about the overall efficiency and effectiveness of this pipeline. It appears to require substantial resources while yielding relatively limited data and coverage.
> > Thus I will maintain my original score.

---

### Official Review · Reviewer_Txic · 2024-11-05

**Soundness:** 2
**Presentation:** 2
**Contribution:** 2
**Rating:** 3
**Confidence:** 3

**Summary:**

This work proposes a AI-based framework that could be applied in generating challenging mathematical questions at scale. Specifically, the framework first extracts mathematical skills from the questions, and generates new questions by pairing up a given skill and a randomly sampled skill (LLMs get involved in estimating the skills being too similar). The questions are filtered by asking LLMs to solve or identify flaws (first tries to estimate the quality from several dimension, then re-solve the questions being qualified), and the filtered questions are passed into human screening. In this manner, a dataset called MATH^2 with questions originated from MATH is collected and a performance law is observed between MATH and MATH^2 in several strong LLMs (performance in MATH^2 is approximately the square of performance on MATH). Besides, it is found that problems in MATH^2 serve as more effective in-context learning examples for solving questions in MATH.

**Strengths:**

An AI-based framework in accelerating the data generation of mathematical reasoning and interesting observation for the performance dependency between the newly generated MATH^2 dataset and the original one.

**Weaknesses:**

Several metrics are missing to better understand the pipeline (e.g. number of questions being filtered out in each stage of the framework, sample skills outlined in the skill extraction stage). Besides, it is not measured whether the the generated questions faithfully integrate the skills mentioned (it is not hard to imagine that for challenging problems, the number of skills involved would exceed two). The qualification of humans getting involved for screening is not introduced. If the entire correctness is still highly depend on the final screening, extending it to the scalable oversight scenarios would still be challenging.

For the experimental results, the explanation of the pseudo square law seems to stand under the assumption that the mathematical skills evolved are orthogonal and of under similar difficulty levels, which may holds for specific datasets. Further evidences are needed to understand the meaning represented by this square-law phenomenon found.

**Questions:**

- I cannot interpret the information in Figure 5.

Typo
- 'creators' in line 319

---

> ### Author Response · Authors · 2024-11-20
> **Response to the reviewer's feedback**
>
> We thank the reviewer for going through our paper and providing helpful reviews. We attempt to answer some of the concerns and questions raised by the reviewer, below.
>
> ### Sample skills outlined in the skill extraction stage:
> We use the same skills as extracted by [1]. We would like to refer you to Table 8 in [1] for the list of skills extracted from the MATH dataset. We will also add this list of skills to the appendix of our paper in the next update (which will be before the end of the discussion period).
>
> ### More metrics for understanding the pipeline:
> Below, we provide some statistics on the number of questions filtered out at different stages of the pipeline for different models. Note that these numbers are representative numbers, calculated on batches of data generated using each model. Questions in the MATH^2 dataset do not all necessarily belong to these batches.
>
> #### Success rate of AI Pipeline
>
> In the table given below, we report the number of questions filtered out during different stages of the AI pipeline (i.e. before the human verification step). **Validation Step** column reports the number of questions that were classified as "invalid" by the models in Step 4 (Question Validation) of the pipeline. **Majority Agreement** column reports the number of questions that were discarded because the final answers resulting from all 4 solution traces in Step 5 (Final Solution) of the pipeline were unique. Additionally, in our pipeline, we use regular expressions to extract the desired output from the rest of the response of the model at each stage. In some cases, the regex failed to extract the desired parts of the model response due to the model not following the specified output format. These numbers are reported in the **Parsing Error** column. The **Total Rejected** column sums the aforementioned columns up and the **Total Generated** column contains the number of questions that were generated in Step 2 (Question Generation) of the AI pipeline.
>
> Overall, GPT-4-Turbo turns out to be the most efficient model, in terms of the number of originally generated questions that made it to the end of the pipeline.
>
> | Model             | Validation Step | Majority Agreement | Parsing Error | Total Rejected | Total Generated | Success Rate |
> |-------------------|-----------------|--------------------|---------------|----------------|-----------------|--------------------|
> | GPT-4o            |        850      |          345       |     48        |       1243     |     1972        |       36.97%       |
> | GPT-4 Turbo       |       1958      |       748          |     64        |        2770    |  5115           |   **45.84%**       |
> | Claude-3 Opus     |       257       |        27          |     24        |    308         |     408         |   24.51%           |
> | Gemini-1.5-Pro    |         935     |     229            |      16       |       1180     |     1434        |    17.71%           |
>
> We plan to add these numbers to the next update of the paper (which will be before the end of the discussion period)
>
> #### Efficiency of Human Verification
> The table given below reports the number of questions annotated per model, and how many questions out of those made it to the dataset. The human annotators were asked to judge whether the questions (after any possible or necessary modifications), were good or a bit too easy. The questions marked "good" were included in the dataset.
>
> | Model          | # of questions annotated | # of questions passed | Success Rate |
> |----------------|--------------------------|-----------------------|--------------|
> | GPT-4o         |     28                   |          3            |      10.71%  |
> | GPT-4 Turbo    |        488               |          116          |      23.77%  |
> | Claude-3 Opus  |      236                 |            51         |       21.61% |
> | Gemini-1.5-Pro |        61                |              40       |  **65.57%**  |
>
> We plan to add these numbers to the next update of the paper (which will be before the end of the discussion period)
>
> #### Cost efficiency of the framework
> Below, we report an estimated cost of the data generation pipeline for each model. For each model, we calculate the average lengths of the input prompts and the generations (summing over all the steps in the pipeline) over 20 interactions. Next, we calculate the average cost for generating 1 question by using the formula:

---

> > ### Author Response · Authors · 2024-11-20
> > **Response Continued**
> >
> > `cost_per_question = avg_input_prompt_length * cost_per_input_token + avg_generation_length * cost_per_output_token`
> >
> > Next, we proceed to calculate the total cost for questions generated by the model using the formula
> >
> > `total_cost = (cost_per_question * num_model_questions_in_math^2) / (human_verification_efficiency * ai_pipeline_efficiency)`
> >
> > where `human_verification_efficiency` and ai_pipeline_efficiency for the given model are taken as calculated in the previous two sections of the response, and `num_model_question_in_math^2` are stated in Section 3.1 of the paper. It is important to note that the result costs would be an estimate of the upper bound, since many of the rejected questions in the AI pipeline stage (see Section **Success Rate of the AI Pipeline**) are rejected in the question validation stage and thus the solutions for such questions are not generated in Final Solution generation stage, saving up on output generation costs.
> >
> > As discussed previously, this cost can be further optimized by using different models at different stages of the pipeline and/or training specialist math problem generation models.
> >
> > | Model               | Avg. Output Prompt Length | Avg. Input Prompt Length |  Cost per Question                                                                                 | Total Cost  |
> > |---------------------|---------------------------|---------------------------|-----------------------------------------------------------------------------------------------|-------------|
> > | **GPT-4-Turbo**     | 4614.85                  | 133833                     | $1.48                                                                   | $1575.60    |
> > | **GPT-4o**          | 6080.95                  | 135618.65                                         | $0.40                                                                         | $30.31      |
> > | **Claude-3 Opus**   | 4066.7                   | 134335.05                                       | $2.32                                                                        | $2233.88    |
> > | **Gemini-1.5-Pro**  | 4851.85                  | 136314.6                                         | $0.23                                                                  | $79.22      |
> >
> >
> > Overall Cost = **$3919.01**
> >
> > We plan to add these numbers to the next update of the paper (which will be before the end of the discussion period)
> >
> >
> > ### Faithful Integration of skills
> > During the human verification stage, the annotators are advised to go through the generated questions and solutions carefully and make sure that a rigorous application of both the skills specified is required to solve the questions. The questions not complying with this requirement are either discarded or modified such that the modified question satisfies the requirement.
> >
> > The square relationship between the performance on MATH^2 and MATH (which was a surprise to us) suggests (as sketched in Section 3.2) that the skills are indeed used in **nontrivial** ways in the questions since on average the new question is much harder for a host of models that were trained in different research groups.
> >
> > ### Qualifications of the Annotators
> > All the annotators involved in the human verification process are graduate students with backgrounds in Computer Science and Mathematics, ranging from undergraduate to graduate levels.
> >
> > ### Pseudo Square Law
> > The reviewer correctly points out that the pseudo-square law follows an orthogonality assumption. Looking at Table 8 in [1], which lists the skills used in our pipeline, one would notice that not all skills, however, are overlapping. Some, albeit few have overlaps. We attribute this to be **one of** the factors contributing to the noise (i.e., deviations from the exact y = x^2 relationship) in the relationship of the performance of models on MATH and MATH^2, as plotted in Figure 3. There are several other factors that may also be thought of as contributing to the noise, including the fact that not all skills are represented equally in both datasets.
> >
> > We are happy to provide any further clarifications and answer any further questions that the reviewer might have.
> >
> > ### Interpreting Figure 5
> > Figure 5 plots the number of times each skill occurs in the MATH^2 dataset (y-axis). X-Axis represents different skills (i.e. each bar is for different skills). The skill occurring the most number of times is `prime_number_theory` - occurring 13 times.
> >
> > We acknowledge that the y-axis label in the plot is misleading. We will replace the label with "Number of Occurrences" in the updated version of the paper.
> >
> > ### Typo
> > Thank you for pointing out the typo. We will fix it in the updated version of the paper.
> >
> > ### References
> >
> > [1] Didolkar et al., 2024. Metacognitive Capabilities of LLMs: An Exploration in Mathematical Problem Solving

---

> > > ### Comment · Reviewer_Txic · 2024-11-28
> > >
> > > Thanks for the response from the authors. Some of the questions are addressed! I believe a more thorough investigation of the pipeline (including pre and post-qualification aspects of the annotators) would make this a stronger contribution (specifically if this would be expected as the ladder for the more challenging problem sets). Therefore, I would keep my score unchanged.

---

> ### Author Response · Authors · 2024-11-29
>
> Dear reviewer,
> Thank you for your comments.
>
> We are not sure if we understand what you mean "pre and post-qualification aspects of the annotators". All the annotators were selected from a pool of graduate students in mathematics and computer science. Could you please elaborate further upon this?
>
> **Stepping stone for more challenging problem sets**
> We completely agree with the reviewer on this and generating even more challenging problems is a very interesting future direction that we have been thinking about. In our experience (as evident from the pipeline proposed), getting frontier LLMs to generate difficult questions by combining even only two skills is not trivial. The models tend to hallucinate a lot while attempting to generate "difficult" questions (It would not be so difficult to have them generate simpler questions that combine two skills). This may be explained by the fact that these questions are meant to be so difficult that the question-generating models themselves may not be able to solve this.
>
> Following the proposed framework, the approach to generating even more difficult questions would be to ask the model to compose k= 3, 4, ... skills at a time. However according to our observations, it turns out to be very difficult for even the frontier models existing currently to do so. Combining k > 2 skills would presumably require even more sophisticated prompting strategies OR training specialist question generation models, for eg. something similar to what is done in [2], but for more difficult questions. Thus, we leave this for future work.
>
> **References**
> [2] Ding et al., 2024. Unleashing Reasoning Capability of LLMs via Scalable Question Synthesis from Scratch.
>
> We would be happy to discuss in further discussion to clarify any further questions that the reviewer might have.

---

> > ### Author Response · Authors · 2024-12-01
> >
> > Dear reviewer,
> > We hope that most of your concerns have been addressed. If so, as the discussion period nears the end, we would appreciate it if you could reconsider your assessment. We’d be happy to engage in further discussions. We would be very happy to engage further in discussions to clarify any remaining concerns that you might have

---

### Author Response · Authors · 2024-11-29
**Manuscript Update**

We thank the reviewers for their useful feedback and suggestions. We would like to bring to your attention that we have updated the paper by incorporating the suggestions. All the modifications are marked in blue. Given below is a summary of the changes made:

* Added example outputs for each step of the AI part of the question generation pipeline in Appendix A.5
* Changed the y-axis label in Figure 5 to "Number of Occurrences"
* Added list of skills extracted from the skill extraction pipeline in Appendix A.4
* Added missing references for models in Section 3.1
* Fixed the typos and added details about the success rate for different steps of the pipeline as well as cost incurred in Appendix A.3.1

We would be happy to engage in further discussions to clarify any concerns and questions that the reviewers have

---

### Meta-Review · Area_Chair_XABm · 2024-12-07

**Metareview:**

This paper proposes a method for generating difficult math questions with AI assistance, based on a set of math skills to be tested extracted from the MATH dataset.  A major driver in the dataset creation pipeline is using pairs of skills to synthesize more complex questions. A question is then generated, solved, validated, and re-solved, all by an LLM. The final questions are screened by humans. Several models are evaluated on the final dataset and it is compared to the original MATH dataset.

The problem of making more challenging datasets for mathematical reasoning is a good one. Researchers in this space may benefit from this dataset.

I see two main drawbacks :

1. KurQ asks about whether MATH^2 is truly out-of-distribution. The performance drop from MATH is not that substantial. High correlation indicates that MATH^2 may not lead to very different conclusions when we go to rank models at their mathematical reasoning capabilities.

2. Lack of generalizability of the pipeline. All four reviewers ask about the role of the humans in the process and how much involvement they have.  Although the paper makes interesting claims about scalable oversight, if we view the pipeline as brainstorming questions which humans then need to validate, it doesn't really provide a viable approach for that.

As a result, I think there are issues with the dataset resource itself as well as the generality of the pipeline as a contribution.

**Additional Comments On Reviewer Discussion:**

Txic asks for metrics to validate the pipeline stages, which are included in the response. The cost numbers are helpful to see. The authors generally provided additional details, but I don't think these clarifications greatly change the impression of the paper. As nXDa says, "I remain somewhat unconvinced about the overall efficiency and effectiveness of this pipeline. It appears to require substantial resources while yielding relatively limited data and coverage."

---

### Decision · Program_Chairs · 2025-01-22

Reject